# Significant Southern Hemisphere contribution to the Indonesian Throughflow over the last 800,000 years

Markus Kienast [1,8], Martina Hollstein [2] ✉, Nadine Lehmann [1,3], Patrick A. Rafter [4], Ziye Li[5], Min-Te Chen [6] & Mahyar Mohtadi [2,7] ✉

The low-latitude flow of water masses from the Pacific to the Indian Ocean, the Indonesian Throughflow (ITF), is a choke point of the surface ocean return flow of the ocean conveyor belt. Even though the significance of the ITF for the modern global ocean circulation and climate has long been established, little is known about the hemispheric origin of the water masses contributing to its overall transport in the past. Here, we take advantage of the distinctly different isotopic composition of subsurface nitrate in the Northern and Southern Hemisphere source waters to document the admixture of these waters in the ITF through time. Our record of bulk sedimentary $\delta^{15}N$ from the Banda Sea, at the heart of the ITF, shows that Southern Hemisphere-sourced subsurface waters contributed significantly to the total ITF transport during the last 800,000 years. Because Southern Ocean processes ultimately set the bio-geochemical source signature of the Southern Hemisphere endmember, the Banda Sea record implies an important conduit by which high southern latitude climate and ocean variability is transmitted into the global ocean.

Water mass transport through the Indonesian Archipelago, the Indonesian Throughflow (ITF), is a critical gauge of salt and heat exchange between the vast Pacific Ocean and the Indian Ocean[1,2]. Specifically, the ITF transports ca. 15 Sv of subsurface-intermediate waters from the Pacific to the Indian, and further into the Atlantic Ocean, where it affects the Atlantic Meridional Overturning Circulation, AMOC[3,4]. As a result, the ITF is a critically important regulator of the global ocean circulation.

Today, the ITF is thought to transport mainly subsurface waters of predominantly North Pacific origin, North Pacific Tropical Water (NPTW) and North Pacific Intermediate Water (NPIW), respectively, with a minor admixture of thermocline and intermediate water of Southern Hemisphere origin, predominantly South Pacific Tropical Water (SPTW) and Subantarctic Mode Water (SAMW), accounting for

20% at most[1,5–7]. These estimates are based on T/S characteristics and velocity measurements[1,8,9], or regional ocean models[6,10]. Actual observations, however, are rather short[2,11] and overwhelmingly focused on the flow through the Makassar Strait[2,12], which is inferred to be almost exclusively of Northern Hemisphere origin. While the significance of the ITF for global ocean heat and freshwater balances has long been established[13], only a few recent studies have explored the nutrient export from the Pacific to the Indian Ocean through the ITF[14,15]. Neither of these studies provides evidence for the hemispheric origin of the Pacific nutrients.

Given its significance in the modern ocean and climate system, a number of studies have investigated changes in the ITF transport on millennial[16–18], orbital[4,19,20], and longer timescales[21]. Although these studies have greatly improved our understanding of past ITF

[1]Department of Oceanography, Dalhousie University, Halifax, NS, Canada. [2]MARUM - Center for Marine Environmental Sciences, University of Bremen, Bremen, Germany. [3]Institute for Marine and Antarctic Studies, University of Tasmania, Hobart, Australia. [4]College of Marine Science, University of South Florida, St. Petersburg, FL, USA. [5]College of Marine Geosciences, Ocean University of China, Qingdao, China. [6]Institute of Earth Sciences, National Taiwan Ocean University, Keelung, Taiwan. [7]Faculty of Geosciences, University of Bremen, Bremen, Germany. [8]Deceased: Markus Kienast. ✉e-mail: mhollstein@marum.de; mmohtadi@marum.de

variations, they remain inconclusive regarding the hemispheric origin of the ITF waters. This uncertainty is partly due to the studies' location in the Makassar Strait[16,18] or the exit passages of the ITF in the Indian Ocean[17,19,22], as well as the reliance on non-conservative water mass tracers, such as ocean surface temperature[4,17,19–21] that are also influenced by changes in atmospheric circulation and greenhouse gas concentrations. In addition, the difference in the T/S characteristics of the Southern and Northern Hemisphere source waters of the ITF[8] is too small to be discernible using paleotemperature proxies. As a result, the relative contributions of Northern and Southern Hemisphere waters to the ITF over time remain uncertain.

The isotopic composition of nitrate, $\delta^{15}N_{(nitrate)}$, clearly distinguishes thermocline waters north and south of the equator in the Western Equatorial Pacific (WEP), with $\delta^{15}N_{(nitrate)}$ differences of more than 3.5‰[23], thus providing an entirely independent tracer to study ITF transport. Southern WEP thermocline nitrate in SPTW is isotopically heavy (7.4–9.4‰) due to the addition of remineralized nitrate at the southern edge of the equatorial upwelling during advection around the subtropical gyre[23,24]. In contrast, NPTW nitrate is isotopically light (~ 5.7 ± 0.2‰) because of remineralization of organic matter with low $\delta^{15}N$ from $N_2$ fixation in the North Pacific gyre[23–25].

Prior studies show that bulk sedimentary $\delta^{15}N$ faithfully records past variations in subsurface/upper thermocline nitrate $\delta^{15}N$ in the WEP[25–27]. Furthermore, there is a good correspondence between the distribution of regional sedimentary $\delta^{15}N$ and expectations based on both model predictions of the isotopic composition of exported marine particulate matter, $\delta^{15}N_{(export)}$[28], and the neural network-based climatology of the isotopic composition of nitrate, $\delta^{15}N_{(nitrate)}$[29], which sets the $\delta^{15}N_{(export)}$ in this region.

This study presents an 800 kyr record of bulk sedimentary $\delta^{15}N$ variability (Supplementary Notes) from the Banda Sea (site MD01-2380, 5°45.64'S/126°54.26'E, 3232 m water depth; Fig. 1a). As the majority of the ITF flows through the Banda Sea (see Supplementary Notes for details), it is ideally situated to monitor the mixed properties of Southern and Northern Hemisphere Pacific contributions to the ITF because it acts as a capacitor[30] and key reservoir[31] of Pacific waters prior to their export to the Indian Ocean. We show that the bulk sedimentary $\delta^{15}N$ variability at site MD01-2380 records an admixture of northern and southern Pacific thermocline waters, with a continuous contribution of isotopically enriched nitrate from the southern WEP thermocline throughout the entire record.

## Results and discussion

Over the last 800 kyrs, $\delta^{15}N$ in the Banda Sea varies between 3.9‰ at ~700 ka and 11.8 ‰ at ~50 ka, with a long-term average of 7.6‰ (Fig. 2b). Sedimentary $\delta^{15}N$ encapsulates the isotopic composition of nitrate in the WEP and at the core site, because nutrient consumption is complete in the oligotrophic WEP[32,33] and central Banda Sea[14]. $\delta^{15}N_{(nitrate)}$ in the Banda Sea is significantly enriched compared to the global average subsurface and deep water ($\delta^{15}N_{(nitrate)}$) of ca. 4.5–5.5‰[29,34]), despite an absence of local or regional water column denitrification in the well-oxygenated[7] Banda Sea.

To address the origin of the isotopically enriched nitrate in the Banda Sea, we compare $\delta^{15}N$ of site MD01-2380 in the Banda Sea to previously published and new $\delta^{15}N$ records from the WEP for the last 25 kyrs (stack records; see the "Methods" section; Fig. 1b and Supplementary Fig. 1), from the northern WEP offshore Mindanao for the last 160 kyr (Supplementary Fig. 2; site MD06-3067), and from the southern WEP offshore Papua New Guinea for the last 800 kyrs (Fig. 2 and Supplementary Fig. 3; site U1486[35]). Our comparison clearly demonstrates that the difference in subsurface $\delta^{15}N_{(nitrate)}$ observed in the water column today[23,25] is coherently captured in sedimentary $\delta^{15}N$ during the last 25 kyrs (Fig. 1b). While there is a large offset (on average 4.3‰) between the sites affected by Northern versus Southern-Hemisphere sourced nitrate, the difference between $\delta^{15}N$ records with the same source of nitrate is minimal (average std. of 0.4‰ and 1.2‰ for the Northern and Southern Hemisphere stack, respectively; Fig. 1b).

Overall, the $\delta^{15}N$ signal in the Banda Sea is clearly offset from the northern WEP stack record, by 2.7‰ on average during the last 25 kyrs, and closer to records reflecting the isotopically enriched Southern Hemisphere influence on $\delta^{15}N$ in the region (offset of 1.6 ‰ on average during the last 25 kyrs; Fig. 1b). The offset between the Southern Hemisphere record at site U1486 and the new Banda Sea record, $\Delta\delta^{15}N$, averages around 2.3‰ for the last 800 kyrs (Fig. 2c). A comparison to the southern WEP stack record (Fig. 1b) suggests that $\Delta\delta^{15}N$ maxima can be attributed to relatively elevated $\delta^{15}N$ at site U1486 compared to

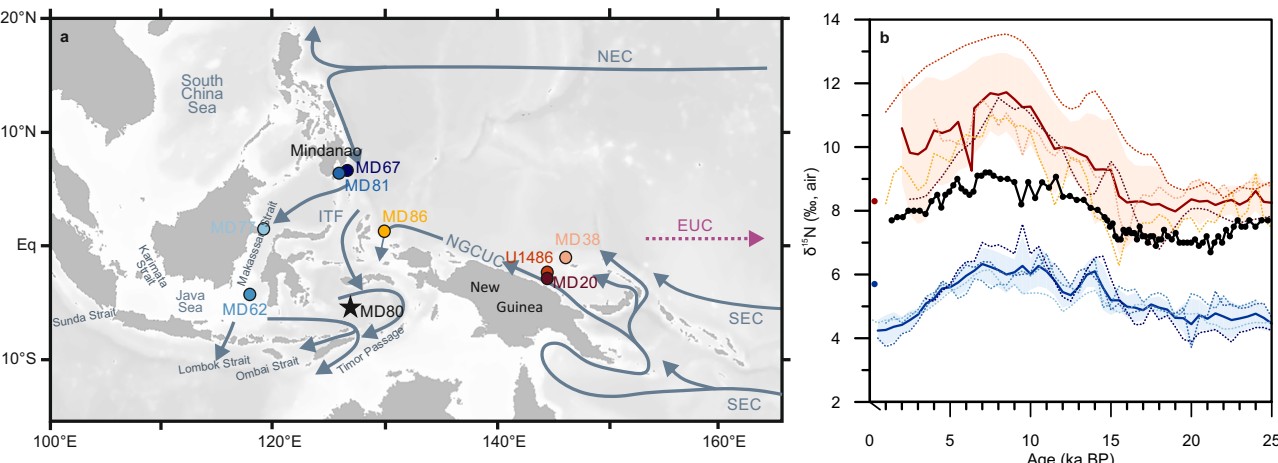

**Fig. 1 | Synthesis of sedimentary $\delta^{15}N$ records from the WEP. a** Bathymetric map of core sites, including site MD01-2380 in the Banda Sea (MD80; black star, this study). Schematic of major subsurface currents adopted from ref. 75: NEC North Equatorial Current, NGCUC New Guinea Coastal Undercurrent, EUC Equatorial Undercurrent, SEC South Equatorial Current, ITF Indonesian Throughflow. **b** Comparison of the Banda Sea $\delta^{15}N$ record (site MD01-2380; black line) to the stacks of records influenced by Northern (blueish tones) and Southern (reddish tones) Hemisphere nitrogen cycling during the last 25,000 yrs. Thick colored lines show the time-dependent mean of four records each, with standard deviations indicated by the colored shadings. See Supplementary Table 1 and Supplementary Fig. 1 for references and details. Dots indicate modern $\delta^{15}N$ of SPTW (red) and NPTW (blue) nitrate[23]. Note that sites are abbreviated to keep the figure clear. The map in (**a**) was generated using ODV[76].

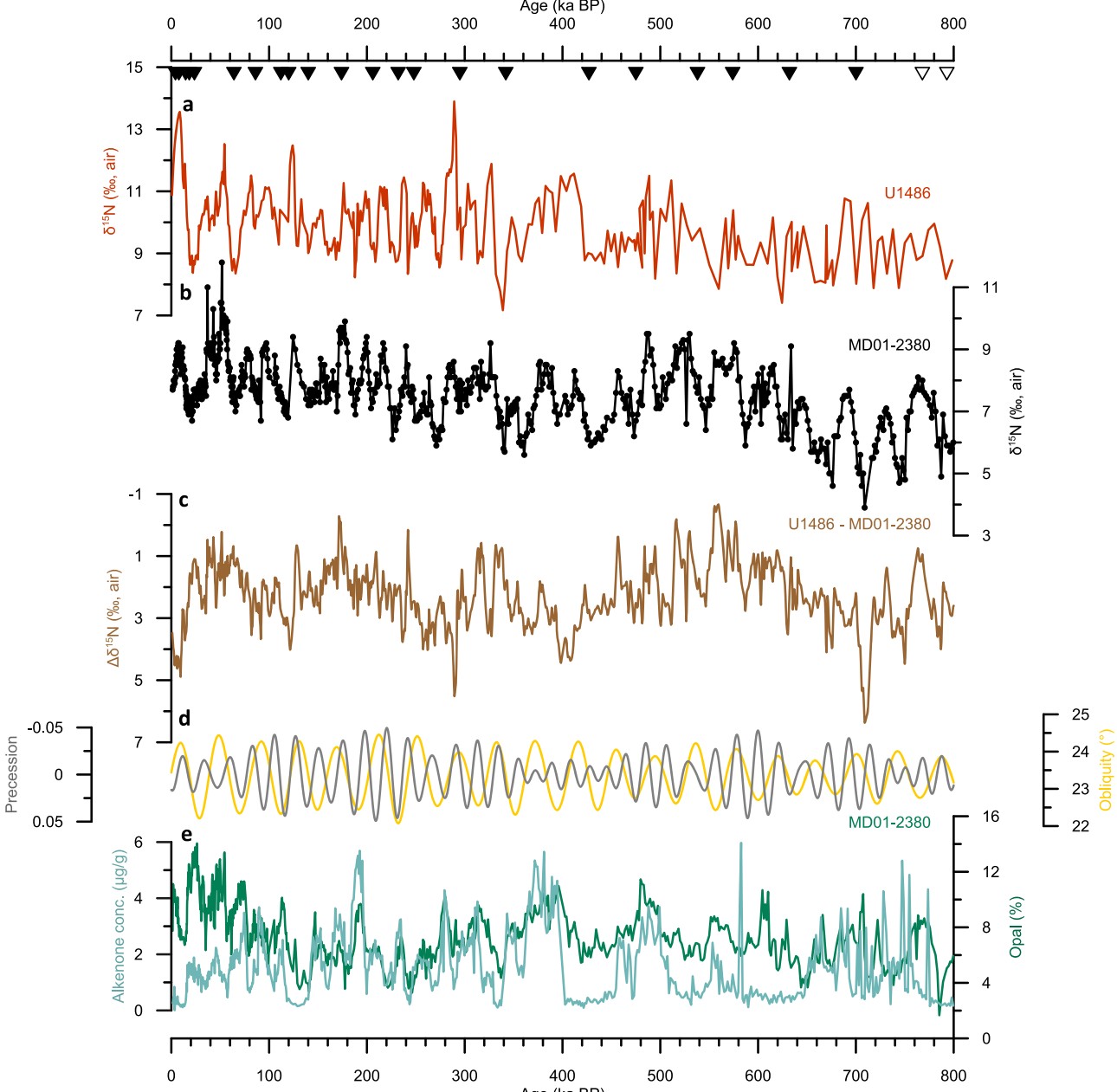

**Fig. 2 | Proxy records with precession and obliquity variability. a** $\delta^{15}N$ record of site U1486[35] (red) and **b** $\delta^{15}N$ record of site MD01-2380 (black), and **c** calculated offset (brown; see the "Methods" section) between the two $\delta^{15}N$ records, $\Delta\delta^{15}N$. **d** Changes in obliquity (yellow) and precession (gray, inverted axis). **e** Percentages of biogenic opal (green) and alkenone concentration (light green) of core MD01-2380. Triangles indicate dating points of MD01-2380.

the other southern WEP records rather than to a significantly reduced Southern Hemisphere contribution. The overall larger spread in $\delta^{15}N$ in the Southern Hemisphere stack, including comparably higher $\delta^{15}N$ values at site U1486 (Fig. 1b), is consistent with modern observations of the $\delta^{15}N_{(nitrate)}$ in the separate branches of the SPTW, reflecting potential variable addition of isotopically heavy remineralized nitrate, diapycnal mixing, and imprints of nitrogen fixation[23].

To provide a quantitative approximation of the Southern Hemisphere contribution to the ITF in the Banda Sea, we used a simplified isotope mass balance, where the $\delta^{15}N$ in the Banda Sea is equated to $\delta^{15}N = x * \delta^{15}N_{North} + (1-x) * \delta^{15}N_{South}$. Using the 160 kyrs $\delta^{15}N$ records of sites MD06-3067 and of U1486 representing the $\delta^{15}N_{North}$ and $\delta^{15}N_{South}$ signals, respectively (Supplementary Fig. 4), the relative fractions ($x$) can be estimated. The Southern Hemisphere contribution ranged from

less than 10% to more than 90%, with an average of slightly above 50% for the past 160 kyrs (Supplementary Fig. 4). The minima can be attributed to minor offsets in the age models between MD06-3067, MD01-2380, and U1486, which cause the records to converge during pronounced $\delta^{15}N$ transitions. To explore the potential magnitude of hemispheric volume transport, we applied the $\delta^{15}N$-derived hemispheric fractions to modern upper-ocean ITF transport using a simple back-of-the-envelope calculation (see the "Methods" section). Considering only the shallow throughflow (0–300 m), our approximation suggests that Southern Hemisphere sources contribute up to 6.8 Sv, with an average of ~2.0 Sv to the ITF outflow through Timor Passage and Ombai Strait (corrected total of ~8.0 Sv, see the "Methods" section) over the past 160 kyrs (Supplementary Fig. 4). However, there are multiple factors that render these estimates an approximation only. (a)

As detailed above, the δ¹⁵N signal at site U1486 is at the high end of the putative Southern Hemisphere δ¹⁵N source. Contribution of the Southern Hemisphere, calculated based on this record, is thus an absolute minimum estimate. (b) Nitrate concentrations in the thermocline today are significantly higher in the Southern compared to the Northern Hemisphere (ca. 8.8 versus ca. 1.9 µM[23]). Our derived volume estimates incorporate this imbalance using nitrate-weighted hemispheric contributions (see the "Methods" section), but this weighting assumes that the difference in modern nutrient concentrations also applies for the past. (c) The ITF involves complex and not fully understood mixing processes, including cooling, freshening, and water mass transformation[2]. (d) A precise quantification of hemispheric contributions to the ITF based on δ¹⁵N would necessitate detailed knowledge of key ocean variables, including nitrate concentrations in the Banda Sea, the ITF outflow, and the main source waters both today and in the past. Despite these uncertainties in the calculation of the relative hemispheric contributions, our results provide unambiguous evidence of a permanent and significant admixture of Southern Hemisphere nitrate to the ITF flowing through the Banda Sea during the past 800 kyrs.

Spectral analysis reveals significant δ¹⁵N variability in the Banda Sea record in the precession and obliquity bands (Fig. 3), with higher δ¹⁵N generally associated with periods of low precession and vice versa (Fig. 2, Supplementary Fig. 5). Assuming that there is a depth stratification to the Northern and Southern Hemisphere source of the ITF waters as inferred from modern T/S observations in the Banda Sea[1], deeper mixing would be tapping into more Southern-Hemisphere sourced waters, resulting in enriched δ¹⁵N(nitrate). Such a deep mixing is rather likely during austral winter and spring, i.e., during periods of low precession[36]. Akin to modern seasonal variability, low precession would result in an overall higher Southern Hemisphere contribution and thus higher δ¹⁵N in the Banda Sea. However, the precessional signal is dominant and of the same sign in δ¹⁵N records of both the Northern and Southern Hemisphere source waters feeding into the ITF (Figs. 3, S3, S6). Therefore, we argue that the precessional variability in sedimentary δ¹⁵N is a direct effect of the equatorial upwelling on δ¹⁵N(nitrate) of thermocline and mode waters of the Southern Hemisphere and the addition of remineralized nitrate[23,24].

Southern Hemisphere intermediate water, specifically SAMW, is considered the main conduit of Southern Ocean nutrients to the upper thermocline and surface layer of the WEP and subsequently the ITF[23,24,37,38]. SAMW-sourced water reaches the ITF upper water column potentially via several pathways. Modern surface ocean radiocarbon data trace the origin of Southern Hemisphere surface water in the Indonesian Sea to the eastern equatorial Pacific upwelling system[37], where SAMW, transported equatorward at intermediate depths, upwells to the surface off Peru[39,40]. SAMW-sourced surface water subsequently flows westward via both South Equatorial Current (SEC) branches, before leaving the Pacific through the ITF and the Banda Sea into the Indian Ocean. While eastern margin surface water may constitute the main component of ITF surface water[37], it likely represents a minor source of nutrients to surface and export production in the WEP and the ITF, as it is largely stripped of its nutrients along its westward trajectory across the Pacific[23,24,41,42]. Modern δ¹⁵N(nitrate) and nutrient distributions in the WEP and the equatorial Pacific instead support the notion that Southern Ocean nutrients resupply the WEP upper thermocline and surface layer via the advection of SPTW, formed through the subduction of nutrient-depleted surface water in the central subtropical Pacific[43,44], and diapycnal mixing with underlying nutrient-rich SAMW as part of the gyre circulation[24]. The Southern Ocean waters do not only feed into the ITF but are also the main source of nitrate in the Equatorial Undercurrent (EUC)[23], which shuttles the precessional signal across the entire equatorial Pacific[24,26] where it is recorded both in the eastern tropical Pacific proper[26,45,46] as well as farther north, at the California Margin (ODP1012[47]; Supplementary Fig. 2). Eventually, the

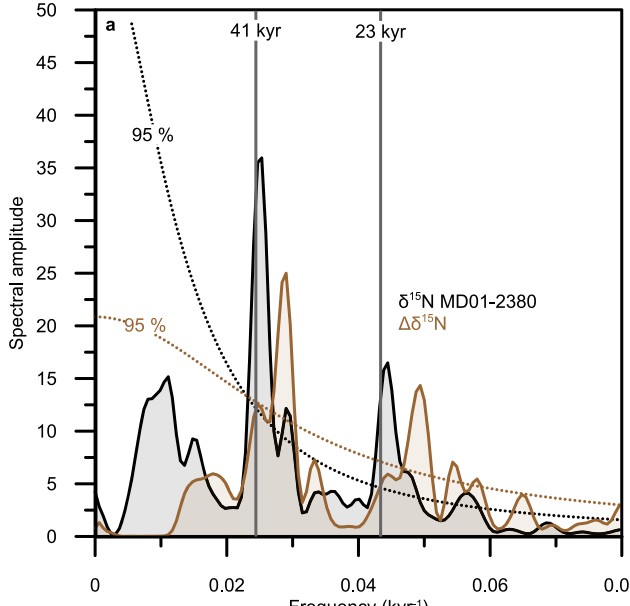

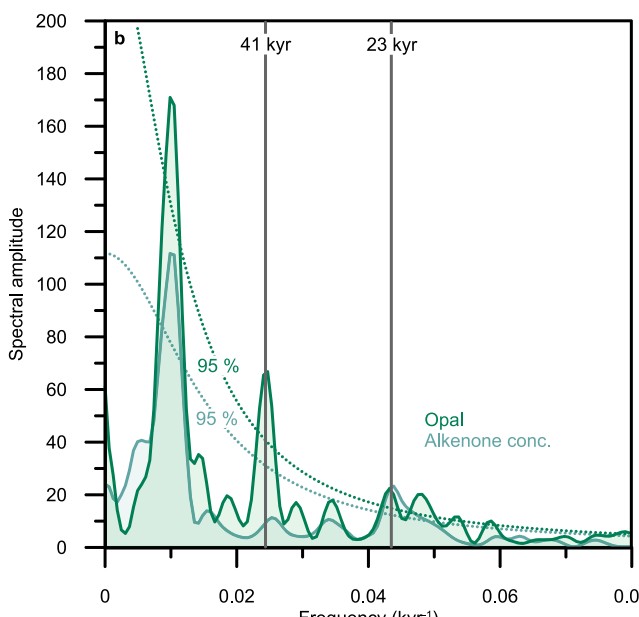

**Fig. 3 | Power spectra of the proxy records. a** Spectra of MD01-2380 δ¹⁵N (black) and U1486 - MD01-2380 Δδ¹⁵N (brown) and of **b** opal content (green) and C37 alkenone concentration (light green) of MD01-2380 calculated with the REDFIT application of ref. 67. Dotted lines denote 95% confidence intervals. Vertical gray lines indicate the frequencies that correspond to precession and obliquity periodicities of 23 and 41 kyr.

Northern Hemisphere WEP nitrate reflects the isotopic composition of nitrate advected from the denitrification zones of the Eastern Tropical North Pacific, overprinted by N₂ fixation in the North Pacific gyre[23,25]. This signal is recorded at site MD06-3067 off Mindanao (Supplementary Fig. 2).

Previous studies have provided evidence for the upwelling of the southern Pacific water masses in the equatorial Pacific during the deglaciations[48,49]. However, to the extent that our scenario to explain the pervasive precessional δ¹⁵N signal of the same sign both north and south of the equator is correct, our records evidence a remarkable long-term stability of the nitrogen cycle all along the equatorial Pacific, and by inference, of the southern Pacific contribution (see also

Supplementary Fig. 2). It is this tight coupling that ultimately explains the precessional signal in both $\delta^{15}N$ and $\Delta\delta^{15}N$ in the Banda Sea.

In addition to the low-latitude precession signal, the $\delta^{15}N$ record in the Banda Sea also displays a strong obliquity signal (Figs. 2 and 3). As discussed in ref. 35, the obliquity signal in the southern WEP is the result of obliquity modulation of both SAMW production and advection, and of sub-Antarctic nutrient utilization. The latter determines the nutrient[50] and isotopic[24] properties of SAMW before it spreads throughout the Southern Hemisphere and provides the nutrients of the equatorial Pacific thermocline. The absence of a strong obliquity signal at site U1486 during the last 800 kyrs[35] (Supplementary Fig. 6) vis-à-vis the clear imprint of an obliquity signal in the Banda Sea (Fig. 3) is plausibly explained by a weaker contribution of SAMW-derived nitrate at the former site, which is also consistent with the overall heaviest, SPTW-derived $\delta^{15}N$ signal at site U1486 compared to the other records from the Southern Hemisphere stack (see above). Considering the limited influence of obliquity modulation on tropical ocean and climate variability, the significant obliquity signal of $\delta^{15}N$ in the Banda Sea provides strong additional evidence of a continuous contribution of high southern-latitude sourced nitrate to the ITF.

New proxy records of past production from the same core in the Banda Sea are consistent with the $\delta^{15}N$-based scenarios developed above. Both alkenone concentration and percent concentrations of biogenic opal (see the "Methods" section) display comparable variability in the precession band throughout much of the record (Fig. 2), suggesting that the dominant mechanism affecting primary production (PP) in the Banda Sea overall is not specific to a certain phytoplankton group. Specifically, higher production in the central Banda Sea is generally associated with high precession (Figs. 2 and 3). Prior reconstructions from the eastern Banda Sea inferred a similar response to precessional forcing, with peak primary production during austral winter[51]. Importantly, the orbital variability observed in the record of biogenic opal (Fig. 3b) provides an entirely independent line of evidence for continuous Southern Hemisphere influence on Banda Sea nutrients, and thus on ITF transport more generally. Sedimentary biogenic opal is a proxy measure of surface ocean production by diatoms[52] (see the "Methods" section), which, in addition to factors affecting all PP, are especially dependent on the availability of silicic acid. Therefore, the significant obliquity forcing evident in only the record of biogenic opal but not alkenones of MD01-2380 (Fig. 3b) indicates continuous Southern Hemisphere influence on the core site, through the Southern Ocean control on thermocline nutrients, in particular silicic acid[50].

No significant 100-kyr cycles can be detected in the $\delta^{15}N$ record of site MD01-2380, despite the potentially strong influence of glacial-interglacial sea level fluctuations on the land–sea configuration along the ITF path. Several studies from the Makassar Strait and the Timor Sea indicate a shift from a surface-dominated flow during glacials to a thermocline-dominated flow during interglacials, driven by sea-level rise and inundation of the Java Sea[19,20]. According to these studies, the relatively fresher waters of the Sunda Shelf/Java Sea induce a freshwater plug south of the Makassar Strait, inhibit a southward flow of surface waters there, and facilitate the thermocline flow of the ITF, akin to its modern seasonal variation[53,54]. This would affect nitrate concentrations, because surface waters are nutrient-depleted, but not the sedimentary $\delta^{15}N$ in the Banda Sea, which is derived from the thermocline and would only change if the relative contribution of the Northern and Southern Hemisphere source waters to the ITF changes. Moreover, the contribution of the Karimata Strait to the ITF is only about 0.8 Sv[2,55]. The Karimata and Lombok Straits are the only shallow passages of the ITF that are affected by changing sea-level on a glacial–interglacial scale and are unrelated to the $\delta^{15}N$ composition in the Banda Sea. The few available studies that suggest sea level-driven changes in the relative contribution of Northern and Southern Hemisphere source waters to the ITF predate our records and are partly related to tectonic reorganizations[21,56].

In conclusion, the record of sedimentary $\delta^{15}N$ from the central Banda Sea presented here provides unambiguous evidence of continued Southern Hemisphere contributions of a substantial fraction of ITF transport during the last 800,000 years. This inference is based on both the overall $\delta^{15}N$ signal vis-à-vis southern and northern hemispheric source signatures and on the strong obliquity signal evident in the records of biogenic opal and sedimentary $\delta^{15}N$. The Southern Hemisphere does, therefore, not only exert a dominant control on modern EUC nutrients[23] feeding the vast equatorial Pacific ecosystem but also provides a substantial fraction of the nutrient flux through the ITF. Given evidence that the ITF contributes significantly to subsurface ocean transport through the Indian Ocean and around the southern tip of Africa overall (Agulhas leakage)[4], and to Indian Ocean nitrogen cycling in particular[57], this inferred far-field transport is thus an avenue to introduce a distinct $\delta^{15}N$ signature, including its orbital modulation, to large parts of the ocean.

## Methods
### Material
Core MD01-2380 was retrieved in the central Banda Sea (5°45.64'S, 126°54.25'E, 3232 m water depth) during the R/V Marion-Dufresne IMAGES VII Cruise[58]. The core was sampled at 5 cm intervals for oxygen and nitrogen isotope analysis, as well as for biogenic opal and alkenone measurements.

To assess the origin of the nitrate in the Banda Sea, we compare $\delta^{15}N$ of site MD01-2380 to a suite of WEP $\delta^{15}N$ records (see Supplementary Table 1) covering the last 25 kyrs, including newly generated records of MD06-3067, MD98-2181, MD98-2177, MD97-2138, MD05-2920, and MD98-2162. These cores were all collected as part of the IMAGES program during the R/V Marion-Dufresne Cruises MD106[59], IMAGES IV, XIII[60], and XIV[61].

We further use $\delta^{15}N$ records from the northern WEP offshore Mindanao for the last 160 kyr (MD06-3067; ref. 25 and this study), and from the southern WEP for the last 800 kyrs (site U1486[35]).

### Isotope measurements
Stable oxygen isotope analysis on planktic foraminifera *Globigerinoides ruber* tests (size fraction 250-355 μm) of MD01-2380 was performed at MARUM, University of Bremen, using a Finnigan MAT 251 mass spectrometer with Kiel I devices. The internal carbonate standard is a Solnhofen Limestone, which is calibrated to the National Bureau of Standards (NBS) 19 standard. The long-term analytical precision is better than ±0.07‰.

Sedimentary $\delta^{15}N$ ($\delta^{15}N_{sample} = [(^{15}N/^{14}N)_{sample}/(^{15}N/^{14}N)_{reference} - 1] *1000$) was analyzed on ca. 60 mg of freeze-dried sediment, homogenized in an agate mortar, packed and enclosed within a tin capsule, placed in a carousel, and combusted to $N_2$ for N isotopic analysis[62]. Analyses were carried out at the Yale Analytical and Stable Isotopic Center YASIC (Yale University), using a Costech ECS 4010 Elemental Analyzer with Conflo III interface (core MD01-2380) and at the Pacific Centre for Isotopic and Geochemical Research (University of British Columbia), using a Carlo Erba NC 2500 elemental analyzer coupled to a Finnigan Mat Delta Plus mass spectrometer (cores MD98-2181, MD06-3067, MD98-2177, MD05-2920, MD98-2162, MD97-2138). The precision of the isotopic analyses based on in-house standard measurements is ±0.2‰.

### Biogenic opal
Biogenic opal of MD01-2380 samples was determined at Dalhousie University by extraction of silica from 20 mg subsamples by a 2 M $Na_2CO_3$ solution at 85 °C, following methods developed by ref. 63. Dissolved silica concentrations in the extract were determined by

molybdenum blue spectrophotometry and multiplied by 2.4 to derive percent opal concentrations. The overall agreement of %biogenic opal with absolute diatom abundances quantified in the same core, but for the last 400 kyrs only supports our interpretation of % biogenic opal as a proxy of production by diatoms in the Banda Sea throughout the last 800 kyrs.

## Alkenone analysis

Alkenone concentrations in MD01-2380 samples were measured at the National Taiwan Ocean University, following the method of ref. 64. Briefly, organic compounds were extracted from ca. 2.5 g of freeze-dried, powdered sediment using a Dionex ASE 350 accelerated solvent extractor with a dichloromethane: methanol (6:4, v/v) solvent at 100 °C and $7.6 \times 10^6$ Pa. Alkenones were separated from total extractable lipids (TEL) using silica column chromatography. Four fractions were collected based on polarity: F1 (n-hexane), F2 (n-hexane: toluene, 3:1), F3 (toluene), and F4 (toluene:methanol, 3:1). For alkenone concentrations, the F3 fraction was analyzed after addition of an internal standard ($C_{36}H_{74}$) using a Hewlett-Packard 6890 Series N Gas Chromatograph (GC) with flame ionization detection (FID). Alkenones were identified by their retention times in comparison to a synthetic standard.

## Age model

The age-depth model of MD01-2380 is based on five calibrated radiocarbon ages, adopted from ref. 65, and on the alignment of the $\delta^{18}O$ record of the planktic foraminifera $G.$ $ruber$ to the $G.$ $ruber$ $\delta^{18}O$ record of U1486[35] (Supplementary Fig. 3). Note that the age model of U1486 was adopted from ref. 35. It is based on a $^{14}C$ date at the core top, on the alignment of x-ray fluorescence data to a nearby core, and on the alignment of benthic $\delta^{18}O$ to the LR04 benthic stack of ref. 66. Because the $\delta^{15}N$ records of MD01-2380 and U1486 show consistent variability over the last 800 kyr, we used these records to refine the age–depth model, in particular toward the bottom of the core.

Prior to calculating the offset between the Banda Sea $\delta^{15}N$ record and the Southern Hemisphere record at site U1486, $\Delta\delta^{15}N$ (U1486 minus MD01-2380), the original records were interpolated to time steps of 2500 yrs.

The age models of cores MD98-2181, MD06-3067, MD98-2177, MD05-2920, MD98-2162, and MD97-2138 were adopted from previous publications as indicated in Supplementary Table 1.

## Spectral analysis

We performed spectral analyses of the $\delta^{15}N$ records of MD01-2380, U1486, and MD06-3067, of the U1486 - MD01-2380 $\Delta\delta^{15}N$ record, and of the alkenone and opal records of MD01-2380 to assess the statistical significance of cyclic variations in the precession and obliquity bands. Prior to the analysis, all records were interpolated to even time steps of 2500 yrs, which is close to the average temporal resolution of U1486. Note that we only used the last 810 kyr of the U1486-record. To exclude long-term variations from the spectral analysis, frequencies below $0.005\,kyr^{-1}$ (for $\delta^{15}N$, alkenone concentrations and opal) and $0.013\,kyr^{-1}$ (for $\Delta\delta^{15}N$) were eliminated from the records, even though they are not significant in the records. The spectral analyses were performed with the REDFIT application[67]. We used a Welch-type spectral window, three WOSA segments, and an oversampling factor of four to increase the resolution of sample frequencies.

## Stack records

The stack records presented here are focused on high-resolution records from the WEP north and south of the equator (see Supplementary Table 1 for details), mostly from identical or close-by sites to water column profiles of $\delta^{15}N_{(nitrate)}$[23]. The separation of cores into a Southern and Northern Hemisphere stack is based on hydrographic consistency, not latitude per se. Thus, cores from Makassar Strait (sites MD98-2162 and MD98-2177), which are inferred to be dominated by northern-sourced waters, are included in the Northern Hemisphere stack despite their southern location. Conversely, the record of ref. 68 from site MD01-2386 is included in the southern stack despite the Northern Hemisphere location. Prior to the calculation of the $\delta^{15}N$ stack records, all time series were linearly interpolated to even time steps of 500 yrs, corresponding to the average resolution of the original records (Supplementary Fig. 1). The Northern and Southern Hemisphere stacks were then calculated as time-dependent mean, i.e. the mean $\delta^{15}N$ of the individual records included in the stacks at each time step, with the corresponding standard deviation. Importantly, the absolute hemispheric $\delta^{15}N$ signals inferred from the stack records, including the association of individual sites with the Northern or Southern Hemisphere, are fully consistent with prior lower resolution records from the more open WEP at large, which are not included in the stack here[26,69,70]. The sedimentary $\delta^{15}N$ of the Northern Hemisphere stack (Fig. 1b, Supplementary Fig. 1) is slightly lower than water column $\delta^{15}N$ nitrate off Mindanao, at the entrance of Makassar Strait[23,25]. This could be due to any combination of two reasons: the water column data only provide a single snapshot in time and are somewhat limited in their spatial and depth resolution of the competing and seasonally varying water masses[71–73] off Mindanao and might thus not be fully representative of minimum $\delta^{15}N_{(nitrate)}$ in the northern WEP. Further, the $\delta^{15}N_{(nitrate)}$ could be made lighter still by additional remineralization of $N_2$ fixation in the Makassar Strait, represented by three of the four records in the stack. Note also, the isotopically most enriched Northern Hemisphere $\delta^{15}N$ record from the open WEP, site MD06-3067, lacks data for the last ca. 3 kyrs[25].

## Estimates of Southern vs. Northern Hemisphere sourced contributions to the ITF volume transport

To approximate the relative Northern vs. Southern Hemisphere contributions to the ITF volume transport, we applied a simplified mixing approach based on $\delta^{15}N$-derived hemispheric source fractions and modern-day ITF volume transport. Because our $\delta^{15}N$ record reflects thermocline nitrate, we used the 0–300 m export through the Ombai Strait and Timor Passage (8.6 Sv[6,53]) as the relevant upper-ocean ITF outflow. We subtracted the net Karimata Strait inflow (-0.6 Sv = -0.8 Sv total inflow minus -0.2 Sv exiting directly through Lombok Strait[10,55]; outflow through the Sunda Strait was not subtracted, because the fraction sourced from the Karimata Strait cannot be quantified) from the ITF outflow, assuming that water transiting through the shallow Karimata Strait is predominantly derived from relatively fresh, nutrient-depleted surface water of the South China Sea. This correction accounts for the notion that nitrate-depleted surface water would impact volume transport and overall nutrient concentrations through mixing while leaving the $\delta^{15}N$ signature largely unaltered. Notably, the Sunda Shelf is exposed during glacial periods. Since our calculation is based on the modern-day ITF transport, this is not considered in our calculations. Since the Karimata inflow builds only a small fraction of the relevant ITF flow, glacial changes would not strongly affect the results.

To convert $\delta^{15}N$-derived hemispheric fractions into volume contributions, we first applied an inverse weighting factor (-4.6×) to the Southern Hemisphere fractions to account for the observed disparity in modern nutrient concentrations between the two endmembers. This weighting ensures that the nutrient-rich southern endmember does not disproportionally affect the inferred volume estimate (i.e., avoids overestimating the contribution of the high-nitrate southern source). The weighted endmember fractions were subsequently normalized to unity and multiplied by the corrected total transport to derive a nitrate-adjusted volume estimate for Southern and Northern thermocline sources to the upper ITF outflow.

## Data availability

All data generated in this study have been deposited in the PANGAEA database under accession code https://doi.org/10.1594/PANGAEA.983489[74]. Data are licensed under CC-BY and accessible without restrictions. The $\delta^{15}N$ data of U1486[35] used in this study are available in the NCEI database of NOAA under accession code https://doi.org/10.25921/4pja-e644.

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

## Acknowledgements

We dedicate this paper to the memory of our first author, Markus Kienast (1969–2025), who passed away after the submission of this manuscript. Markus was an exceptional scientist, mentor, colleague, and dear friend whose intellectual rigor, generosity, and candor profoundly shaped this work. His enthusiasm for science and his joy in collaboration left a lasting imprint on us and on all who had the privilege to work with him. He is deeply missed. The authors thank Brad Erkkila (Yale Analytical and Stable Isotope Center, YASIC) for $\delta^{15}N$ analysis and Emma Taniguchi (Dalhousie) for sample preparations and determinations of biogenic opal. Birgit Meyer-Schack, Maike Steinkamp, and Henning Kuhnert at the Laboratory for Stable Isotopes (MARUM) are acknowledged for $\delta^{18}O$ analysis. Lowell Stott (USC) kindly shared an unpublished age model of core MD98-2177. This study would not have been possible without sediment core material recovered during *Marion Dufresne* (IMAGES) and RV *SONNE* expeditions. We thank the chief scientists, captains, and crews of these sampling campaigns for their dedication and efforts, and Luc Beaufort, Lowell Stott, and the OSU Core Repository for sharing sample material. We gratefully acknowledge funding support by NSERC (DG and SRO programs) to M.K., the German Research Foundation, DFG, projects BASE-ITF (MO 2546/3-1) and Cluster of Excellence "The Ocean Floor—Earth's Uncharted Interface", EXC 2077, 390741603 to M.M., NSF China (42006057) and the China Postdoctoral Council Fund (PC2019086) to Z.L., Taiwan MOST and NSTC grants (MOST 110-2116-M-019-005, MOST 111-2116-M-019-004, NSTC 112-2116-M-019-004) to M.-T.C., and the Taiwan (M.-T.C.) and Canada (M.K.) IMAGES programs.

## Author contributions

M.K.: Writing—original draft, writing—review and editing, formal analysis, funding acquisition, and conceptualization. M.H.: Writing—original draft, writing—review and editing, formal analysis, and visualization. N.L.: Writing— review and editing, and formal analysis. P.A.R.: Writing—review and editing. Z.L.: Formal analysis. M.-T.C.: Formal analysis and funding acquisition. M.M.: Writing—review and editing, funding acquisition, and conceptualization.

## Funding

## Competing interests

The authors declare no competing interests.
