## [Transparent Peer Review file · Nature Communications]

Significant Southern Hemisphere contribution to the Indonesian Throughflow over the last 800,000 years

Corresponding Author: Dr Mahyar Mohtadi

Version 0:

Reviewer comments:

Reviewer #1

(Remarks to the Author)

In this paper, Kienast et al. use the $\delta^{15}\text{N}$ of the bulk organic matter in eight sediment cores to estimate 1) the relative amounts of Northern and Southern Hemisphere water in the Indonesian Throughflow (ITF) and 2) the way that the sources vary over time. Kienast et al. find that Southern Hemisphere water makes up more than half the ITF most of the time. This is a very interesting and compelling result. Given the longstanding controversy regarding the sources of the ITF, it is somewhat surprising that this simple analysis was not done years ago.

The analysis in the paper is very straightforward, and I am inclined to say that the paper could be published almost as is. An important clarification is needed, however. The ITF is not a single entity. It is spread over 1500 m or so and the water near the top is quite different from the water near the bottom. The part of the ITF that is examined in this paper is the near-surface and upper thermocline parts of the flow. This is acknowledged in the paper but the consequences are not.

The basis for a Southern Hemisphere source for the ITF was laid out by Stuart Godfrey in 1989. Basically, the westerly winds blowing along the southern boundary of the Pacific basin at $\sim 45^\circ\text{S}$ push a large volume of water northward into the basin. If there is a gap along the western boundary of the basin, as there is in the Indonesian Seas, the water that is pushed across 45°S exits through the gap. Godfrey predicted that the flow through the gap should be about 15 Sv -- an astonishingly good prediction.

The water that actually flows through the Indonesian Seas, however, seems to have a northern source, as had been pointed out a few years earlier by Arnold Gordon in 1986. Gordon thought that a fairly large volume of old NADW upwells in the North Pacific and returns to the North Atlantic via the ITF and Agulhas eddies. Gordon saw the Throughflow as part of the overturning of NADW. These two ideas about the Throughflow couldn't be more different.

Kienast et al.'s results, taken at face value, tilt toward the Godfrey's approach. The problem is that Godfrey, in his original paper, was not the least bit concerned with the properties of the water that flows into the Pacific across 45°S . Kienast et al. need to be concerned about these properties because the properties of the water that flows across 45°S are very different from the properties that are monitored by their sediment cores in the western Pacific.

The water that enters the Pacific via the winds along $45\text{--}50^\circ\text{S}$ is most readily characterized as Subantarctic Mode Water (SAMW). SAMW is roughly a 50:50 mix of old deep water that is drawn up to the surface south of the ACC and subtropical thermocline water from the South Pacific. The old deep water in SAMW is a signal that the ITF is a hybrid flow, some of the ITF is wind-driven, some is part of the ocean's overturning. Most importantly, SAMW is quite cool (because of where it forms). Kienast et al.'s near-surface upper thermocline southern source, in contrast, is quite warm. So, if Kienast et al. are correct, a major water mass transformation must be taking place along the path that SAMW takes from 45°S to the Indonesian Sea.

I published a pair of papers in 2019 (with co-authors Ellen Druffell, Robert Key, and Eric Galbraith) that examined this transformation (Toggweiler et al., *Upwelling in the Ocean Basins North of the ACC*, Parts 1 and 2, *JGR-Oceans*, 10.1029/2018JC014794 and 10.1029/2018JC014795). We used the surface D_{14}C to monitor the circulation. D_{14}C is similar to $\delta^{15}\text{N}$ in the sense that both tracers are weakly modified by contact with the atmosphere. Our results showed that much of the southern water that enters the Pacific across 45°S is upwelled to the surface off Peru. We were also able to

assess the surface waters that enter the Indonesian Seas, and we found that the southern water from Peru is the main component. Our results are thus quite consistent with the results in the present paper. The latter finding is discussed on page 14 in the Supporting Information for Part 1.

In Part 2, we simulated the surface distribution of D14C in a modern climate model and found that the none of the subantarctic water from 45 deg. S in the model was reaching the surface off Peru. It tended to exit the Pacific as a cool subsurface flow instead. Thus, the inflow across 45 deg. S in the model remained unmodified as it made its way to the Indonesian Seas. This discrepancy is discussed on pages 10-13 in Part 2. The discrepancy then led us to a hypothesis to explain how so much SAMW manages to upwell off Peru in the real world.

Our hypothesis is that the formation of NADW in the North Atlantic draws away the warm buoyant water that is piled up in the western Pacific by the easterly winds in the equatorial zone. The volume of warm buoyant water that is drawn away in the west thereby determines the volume of cool water that is upwelled to the surface off Peru. Our model did not manage to draw any of the piled up water in the west away, so there was much less upwelling off Peru in the model than observed. As a result, the inflow across 45 deg. S in the model ends up leaving the Pacific as a cool subsurface flow.

The reason that I mention all this is that our model ended up generating a circulation that is somewhat like the one that Godfrey described, i.e., the water that leaves the Pacific via the ITF is largely the same water that enters the Pacific across 45 deg. S. If the real ocean worked this way, Kienast et al. would not see any evidence for a southern source in their sediment cores. (The southern outflow would be too deep to influence the organisms that produce the organic matter that is extracted from the sediments.)

So, IMHO, Kienast et al. are able to "see" the southern source in their cores because the inflow across 45 deg. S is transformed off Peru. The paper would be greatly improved if it included this perspective.

J. R. Toggweiler

Reviewer #2

(Remarks to the Author)

Review

General Written Evaluation

First of all, I want to apologise for taking longer than expected to complete this review. While dangling the potential of a significant scientific paradigm shift in front of the reader, this manuscript woefully falls short in transporting this message in its current structure. It was thus difficult to review, as it presents excellent data but wraps it in a somewhat problematic narrative that often fails to establish key concepts early enough or downright misrepresents large portions of a large body of especially more recent scientific literature dealing with the same study area.

The manuscript titled 'Significant Southern Hemisphere contribution to the Indonesian Throughflow over the last 800,000 years' tries (but has failed to convince me) of the fact that previously a critical part of the Indonesian Throughflow water mass contribution has been completely ignored in previous literature. Namely, they tout the hypothesis that the southern hemisphere water contribution to ITF waters is completely ignored in the literature and that their dataset is the first to allow proper evaluation of said oversight. Here, I wish to emphasise that their data shows compelling evidence for the southern hemisphere water influx if taken at face value. I simply disagree with how they tried to discredit the rest of the scientific community to generate an overinflated impact of their data – and I know this is probably done primarily to fit the "flashy" style of many articles that are published in "high-impact" journals such as Nature Communication. A trend, I feel, is wholly unnecessary if one can let compelling data speak for itself.

With that (admittedly very harsh) opening statement, I wish to take a short step back and explain why I have arrived at this overall negative conclusion after several re-reads of the text and despite a strong motivation to like this manuscript. Despite all the negativity, I still think the author's data is valid and valuable. But – and this is crucial – the authors did not manage to be convincing in framing this data and connecting it to the overall significance of the region. Even more problematically, they also did not do their due diligence in performing even rudimentary tests to ascertain their proxy assumptions are valid – they cite data and rattle off reasons the reader is supposed to believe (disclaimer: I do 'believe' them, but only by the benefit of being very somewhat familiar with the data basis and the literature). I, therefore, wish to fully emphasise that I do not make this statement lightly, especially considering that many of the authors are well-regarded experts in the field and have even (co-)authored several papers dealing with such proxy validation. I am baffled by the lack of due diligence regarding data evaluation.

The fact that the authors do not even bother to include such fundamental principles in their methodology or their supplements is not adequate for any reputable scientific journal, let alone a journal intended for a general scientific audience such as Nature Communications (although they do touch upon it in the discussion chapter). I thus feel that this manuscript needs a lot of additional work on the drawing board before it can be considered for publication in a scientific journal. If the authors wish to publish it in a journal geared towards a general scientific readership (such as Nature Communications), they will also need to critically re-evaluate some of their visual style choices, as well as restructure their frame story and writing clarity. I found their frame story lacklustre in its impact, the beginning (not necessarily the latter half) of their data discussion distracting, and their figures confusing and challenging to read.

Below, I have illustrated the basis of this conclusion through a set of general chapter-specific comments and specific line-by-line comments throughout the manuscript. I am sorry I cannot be more positive in my assessment of this work, but the overall presentation of the data does not live up to the impact the authors wish it to have—or that the data itself would deserve.

Chapter-specific Comments

Abstract

The abstract is scientifically imprecise and overemphasises a narrow viewpoint of an unknown to inflate the study's significance. Such imprecision should be remedied, as ample work on the region's modern oceanography adequately describes the ITF system's complexity.

Introduction

I am somewhat worried about how the authors push an agenda and narrative within this introduction that often critically omits the nuance of previous works. This generates a (perceived) problem in the literature that the authors now set out to solve.

This, I feel, detracts from the actual importance and validity of the author's dataset (albeit some methodological misgivings I may have with it; see below)

Results and Discussion

Overall, the data presentation is lacklustre. Figure 1 is not labelled correctly. The erratic style choices in how data is visually represented make this manuscript completely unreadable, especially for people with colour-impaired vision. The authors will need to fix both the clarity of their discussion and the clarity of data visualisation before any (more critical) discussion of their data in terms of scientific interpretation can be started.

Furthermore, considering that this study spans the last 800 kyr, I would have appreciated at least some discussion on the effect of sea level on varying ITF connectivity. The Macassar straight, especially, is well known to have had significantly altered connectivity during the late Pleistocene glacials. The authors ascribe an "over-emphasis" of the Macassar Straights' contribution to the story but fail to account for sea level-driven ITF changes entirely. This is a critical problem that needs to be addressed, especially since they do cite some Plio-Pleistocene studies that touch upon these changes in the context of ITF dynamics (e.g. their ref18).

Finally, the authors offer a very well-crafted discussion of their data and manage to place it into a large global earth system context in the latter half of their results and discussion chapter. However, this discussion makes it even more difficult to accept the statement made in their introduction, as they offer numerous citations to again prove that the connectivity and contribution of southern hemisphere intermediate waters in the tropics is well known – I cannot get over the fact that the authors decided to eschew the (undeniable!) importance and impact of their data in such a way, that it partly and erroneously represents the large and growing body of literature dealing with the Pleistocene to recent dynamics of the Indonesian Archipelago.

I am very disappointed that the authors discuss precession and obliquity patterns but fail to contextualise these changes in terms of large-scale glacial-interglacial variability – especially sea level-related restriction of large portions of shallow pater ITF connectivity. This would better support their (not necessarily new, but certainly very well supported by data) ideas regarding the Indonesian Archipelago's role as a key "choke point" in the global thermohaline overturning circulation and thus also nutrient fluxes in the ocean. I am also disappointed that the authors fail to emphasise the importance of direct pathways of AAIW/SAMW into the lower latitudes and the northern hemisphere and how these intermediate water pathways directly influence upwelling in the open ocean compared to their data from the enclosed Banda Sea. – that being said, I am well aware of the lack of congruency between their data and G-IG forcing, both in the spectral and the visual domains. That, however, does not mean the authors can skip discussing it.

In conclusion, I do not feel this manuscript is ready for publication despite the authors' undeniable data quality and considerable effort in data synthesis. The text, figures, and discussion are simply not on a level that reflects both minimal quality standards and the state-of-the-art of the current literature. This is especially problematic, as even the authors' starting premise does not reflect the general understanding of modern-day oceanography of physical and chemical oceanographers or the paleoclimate community (based on my experiences).

Methods

I am very concerned that the authors do not evaluate the source of their organic matter further. Generally applying the TOC:TN ratio and $\delta^{13}\text{C}_{\text{org}}$ should have been measured and evaluated to show the marine origin of the bulk OM they measured. Please follow the recommendations of Meyers (1994) here:

Meyers, P. A. (1994). Preservation of elemental and isotopic source identification of sedimentary organic matter. *Chemical Geology*, 114(3–4), 289–302. [https://doi.org/10.1016/0009-2541\(94\)90059-0](https://doi.org/10.1016/0009-2541(94)90059-0)

Before the authors ascertain the marine origin of the presented bulk organic matter $\delta^{15}\text{N}$ record, I cannot allow this work to be published in good conscience. Again, I do not doubt the validity of the data; I am just baffled by the authors' lack of due diligence in terms of method justification here.

Supplements

I am putting this subchapter here in the hopes that the authors, for their next attempt at submission, at least will consider adding a much more extensive supplementary section that can better justify their arguments. While short-form papers are excellent at succinctly conveying compelling science stories/news, the meat, data/interpretative justification, and extensive hypotheses testing should always be present somewhere – usually in the supplements. Considering that the authors purport to present such a paradigm shift in this manuscript, I am consequently missing a systematic and extensive discussion of these concepts in a broader context. Hence, why are there no supplements except four figures, which could each have easily been integrated as part of one of the main figures? Fig. S1 could be part of Fig. 1; Fig. S2 could be part of Fig. 2, and so on...

Line-specific Comments

Abstract

Line 3: Please do not use iconographic in this context. It has a narrowly defined meaning, and although it can be used in this context, it may be not very clear to general readers.

Line 6-8: I wholeheartedly disagree with that statement. Major sources of subsurface (lower thermocline) water masses flowing through the ITF have always been attributed to the Pacific South Equatorial Current. Everybody with a basic understanding of physical oceanography will grasp that concept. This amount of imprecision is unacceptable in a scientific

abstract. To quote from

Introduction:

Line 26: I am somewhat worried the authors may not have fully grasped the information they referenced [6]. It has always been known that southern hemisphere waters contribute appreciably to the ITF that enters Halmahera (especially in the geologic past!). A more detailed reading of the literature will also reveal that the most significant part of the southern hemisphere sources water flow at lower thermocline and sub-thermocline depths.

Here are some suggestions for further reading:

Gordon, A. L., & McClean, J. L. (1999). Thermohaline Stratification of the Indonesian Seas: Model and Observations*. *Journal of Physical Oceanography*, 29(2), 198–216. [https://doi.org/10.1175/1520-0485\(1999\)029<0198:tsofis>2.0.co;2](https://doi.org/10.1175/1520-0485(1999)029<0198:tsofis>2.0.co;2)
Tillinger, D. (2011). Physical oceanography of the present-day Indonesian Throughflow. Geological Society, London, Special Publications, 355(1), 267–281. <https://doi.org/10.1144/sp355.13>
Feng, M., Zhang, N., Liu, Q., & Wijffels, S. (2018). The Indonesian throughflow, its variability and centennial change. *Geoscience Letters*, 5(1), 3. <https://doi.org/10.1186/s40562-018-0102-2>

Line 34 – 41: This is a gross oversimplification of the literature cited and a limited view of the large body of works available. I do not feel that this is how a scientific argument should be built (irrespective of whether I agree with its validity, I most certainly do not). Regardless of my misgivings in how this received issue is framed, I am most baffled by how the authors can cite citations 15-18 in support of such an argument. Finally, I find the term 'tacitly' somewhat inflammatory here. It seems to be placed with the sole intention of verbally enforcing a scientific gap that, in all actuality, does not even exist. Furthermore, it becomes clear already by this portion of the manuscript that some key aspects in the behaviour and mixing of ITF water masses in the Indonesian Archipelago may not have been considered fully by the authors.

Line 41 – 45: I agree with that statement. However, I am not happy with the author's attempt to bend the evidence of physical oceanographic observations to fit their narrative, which is pervasive in their introduction.

Line 46 – 52: Yes, that is true, but the authors do not measure nitrate in their study. They measure bulk sedimentary organic matter nitrogen isotope ratios. Therefore, local denitrification and nitrogen recycling of the sediment, as well as riverine influx of terrestrial nitrogen sources such as land plant-derived organic matter from the Indonesian Archipelago, will need to be considered in more detail to substantiate proxy claims the authors put forward in this manuscript.

Line 53 – 58: All except one (Citation 26), all citations supporting this argument have been authored or co-authored by one or more of the authors of this manuscript. While I do not doubt the validity of their previous studies, it may indeed explain some of my earlier misgivings concerning the tone of the introduction so far.

Results and Discussion

Line 68 – 88: I have read these paragraphs multiple times, and despite being extensively familiar with the study region in terms of modern oceanography, IODP, and MD site locations, this was more than just hard to parse. The conclusion here is thus: A more general reader simply interested in the core message of this work will be lost before any meaningful discussion has commenced. Please revise for clarity. This includes better visualisation of the site location (colour difference is not enough!)

Line 73 – 76: Yes, very curious. I would love to see more discussion on this fact. Possibly focused on (discounting) the possible contribution of terrestrial organic matter $\delta^{15}\text{N}$ from the not insignificant riverine influx in the Indonesian Archipelago and also north in the South China Sea, including the Mekong and Yellow Rivers.

Line 89-101: This paragraph finally, for the first time in the manuscript, provides a neutral and easy-to-follow discussion on the different sources and their potential contributions. However, as stated above, I am still missing a text of the null hypothesis to disregard the local influx of organic matter into the Banda Sea (disclaimer: I do not doubt that the data and interpretation of the authors will most likely remain valid, but it still needs to be done in the interest of due diligence of data analyses).

Line 116 – 117: This is a pretty offhanded sentence. The authors should at least reference (or better yet include) such a mass balance model if they mention it!

Line 126 – 131: I am not considering precessional-driven climate changes in the tropics. The authors immediately move to water mass stratification but ignore the intertropical convergence zone and the interlinked monsoonal changes in this discussion. Again, they need to confidently discount this to substantiate the presented hypotheses.

Line 140 – 148: don't doubt precessional control over equatorial upwelling in the Pacific. But I am beginning to doubt that the authors have considered all the nuances underpinning this system, especially in the context of seasonally shifting Hadley circulation. Also, in the end, they mention ODP Site 1012. Would it be possible to show this on a map and, ideally, graphically illustrate the connection to the discussed dataset? This would help readers understand the argument the authors are trying to make here.

Line 146: EUC has never been defined. Please do so.

Line 148 – 156: This brings back the discussion nicely, but I am still a bit confounded that the authors show the close comparison in ODP Site 1012 and Site MD06-3067 but fail to display this connectivity accurately in Figure 1. This detracts

from the quite powerful statement they make here.

Line 159 – 167: I have to commend the authors here. This is a well-crafted explanation and reasoning. It reflects the state of the art of our understanding and presents how their data fits into it. However, I still fail to see how their data constitutes a paradigm shift on the level they hint at in their introduction.

Line 170 – 184: First: This would be better suited in the methodology and/or earlier in the discussion stage. Second, I feel this is an inadequate way of assessing data validity, as it hinges on the "because we say so" approach. No concrete testing is applied.

Line 197-204: Yes, but that is also true for any other upwelling region globally, so please be more specific on how it supports their line of argument regarding the uniqueness of their $\delta^{15}\text{N}$ data. Biogenic opal essentially shows the same after all.

Line 209 – 214: This is very weakly developed regarding scientific arguments. The authors ignore that the Indian Ocean has a direct (and I would argue very well understood) connectivity to the southern ocean. This connection has been extensively studied on different time scales, as it plays a crucial role in controlling productivity patterns in the Arabian Sea. Describing the contribution of the ITF to the overall $\delta^{15}\text{N}$ cycle of the Indian Ocean is undoubtedly important, but again, in terms of my understanding, nothing 'new' in the sense of a glaring gap of understanding. Irrespective of the above statement, their results still constitute the first data set I have seen that convincingly and compellingly shows this connection and its variability over the last 800 kyr. Thus, at the end of my three read-throughs of their manuscript, I am still immensely disappointed by the author's presentation of the significance of this enticing dataset. The lack of willingness to contextualise the complexity within the framework of existing literature is especially baffling, and to any knowledgeable reader, it will be similarly obvious and jarring.

Reviewer #3

(Remarks to the Author)

The study investigates the relative contribution of North and South Pacific in terms of sourcing nitrate to the Banda Sea over the past 800,000 years using nitrate isotopes. Subsequently, these findings are used to infer the fractional contribution of the South Pacific water on the Indonesian Throughflow (ITF) into the Indian Ocean. The study is well-written, and of significant interest to the broader oceanographic community; since it contributes to our understanding of water composition and nutrient transport associated with the Indonesian Throughflow, and the resulting effects on nutrient availability and distribution across the Indian Ocean and on a global scale. The scope of the study and its findings are well-suited for publication in *Nature Communications*. In my opinion, the methods are well explained, robust and support well the authors' arguments; although I want to highlight that I am not an expert in isotope analysis. However, I do have some minor queries mainly concerning the representativeness of the Banda Sea for the entire ITF. Hence, I recommend some minor revisions as described below prior to publication.

1. Banda Sea and ITF: Throughout the study, the ITF transport and the water passing through the Banda Sea are treated as equivalent. While a significant portion of the ITF outflow through the Ombai and Timor Straits does indeed flow through the Banda Sea, this simplification is not entirely accurate. Observations indicate that approximately 20% of the ITF transits through the shallow Lombok Strait, directly downstream from Makassar and before the flow turns eastward towards the Banda Sea. Therefore, the maximum proportion of the ITF that can be explained by examining the Banda Sea would be 80%. Moreover, recent high-resolution models (Guo et al., 2023) suggest that some water from the Ombai Strait and along the Nusa Tenggara region exits directly into the Indian Ocean before move eastwards towards the Banda Sea, which further reduces this percentage. Additionally, the contribution from the Lifamatola Passage to the Ombai Strait (30% of the total ITF) appears to primarily follow a path along the western boundary of the Banda Sea, rather than traversing its interior. Hence, in my opinion, inferences made from observations within the interior of the Banda Sea are most applicable to the portion of the ITF flowing through the Timor Sea, which accounts for approximately 50% of the total ITF. I suggest that you explicitly highlight and clarify this distinction throughout the text, particularly in the Introduction and conclusions. Specifically, that your discussion regarding the contributions of South and North Pacific sources to nutrient transport relates to this Timor Sea component of the ITF (or at least a portion of the ITF), rather than the entire ITF.

Guo Y, Li Y, Yang D, Li Y, Wang F and Gao G (2023) Water sources of the Lombok, Ombai and Timor outflows of the Indonesian throughflow. *Front. Mar. Sci.* 10:1326048. doi: 10.3389/fmars.2023.1326048

2. Figure 1. To help readers, particularly those unfamiliar with the Indonesian Throughflow, I recommend illustrating the complete ITF pathway in your schematic depiction. Specifically, in Figure 1a, please include the outflows through the Lombok Strait and the Ombai Strait, in addition to the existing outflow through the Timor Sea. This will provide a more comprehensive visual representation.

3. Lines 102-108, Figures 1b and S1, and use of U1486 to represent the $\delta^{15}\text{N}_{\text{South}}$: U1486 (purple) exhibits significantly different and larger values (lines 96-98) compared to the other 'southern' profiles, falling outside their standard deviation. So I am not sure why you used this record-site to characterise and estimate the fractional contribution of southern vs northern water sources in your equation in line 104 and Figure S4. Maybe I am missing something or I have misunderstood but could you please clarify for this selection. Additionally, could you please provide an alternative estimate of the fractional contribution using a different southern record, or perhaps the mean of these records. Given the linear combination of $\delta^{15}\text{N}_{\text{North}}$ and $\delta^{15}\text{N}_{\text{South}}$ in the equation in line 104, using the mean and standard deviation of both northern and southern records could provide a mean fractional contribution estimate with a corresponding standard

deviation. In my opinion, this approach will strengthen your argument and reduce its dependence on the choice of specific record-sites.

4. Depth of records: This probably stems from my ignorance in terms of using delta¹⁵N, but I was wondering whether the results are sensitive to the water depth of these records. Does the U1486 shallower depth contribute to its observed differences and higher delta¹⁵N than the rest of the southern records. Similarly, does the considerably deeper depth of MD01-2380 (3232 m) relative to both the northern and southern records affect the overall findings. Given the broader interest in your results, a brief explanation of how water depth might influence delta¹⁵N analysis and your results (if it does influence them) would be beneficial, particularly for readers less familiar with isotope analysis.

Version 1:

Reviewer comments:

Reviewer #3

(Remarks to the Author)

As I highlighted in my review of the previous version of the manuscript, the study is of interest to the wider oceanographic community as it contributes to improving our understanding of the Indonesian Throughflow and associated water/nutrient transport into the Indian Ocean. The study's conclusions and authors' arguments are well supported by the analysis and methodology. My previous minor comments have been addressed in the revised version of the manuscript, and I recommend the manuscript be accepted for publication as is.

Reviewer #4

(Remarks to the Author)

Manuscript Review for
Significant Southern Hemisphere contribution to the Indonesian Throughflow over the last 800,000 years
Submitted by
Kienast et al.

Overall remarks

I have been asked to review this manuscript at a time, where a rebuttal already answered to comments from three reviewers. In my review, I will provide my own comments, and include my thoughts on previous comments and their responses in the rebuttal.

Kienast et al. provide new, exciting paleoclimate data which use a unique approach to distinguish northern vs southern water mass sources at their site. I am excited by the nitrogen isotopic approach on multiple cores, and support the publication in Nature Communications, but I have several major points that need clarification before I can support the publication of the manuscript. These include

(a) The current methods section is not well organised and does not include sufficient information on the methodology of the work, especially for a journal like Nature Communications that reaches a wide audience. This makes it hard for any reader that does not have a geochemical background to judge whether the assumptions made in this manuscript are reasonable or not. First, it appears that the authors measured d¹⁵N on two cores MD01-2380 and U1486, but the methods only describe steps taken for MD01-2380. Second, the method section "Isotope measurements" does not reference any methodological studies, hence I assume this method is not yet published. In that case, I would expect a more in detail description of how the sedimentary d¹⁵N was collected. For example, after homogenisation, how did the samples enter the mass spec? Third, the chapter references a row of cores that also provided samples for bulk d¹⁵N analysis, but these cores are not really referred to in the main manuscript. More information helping the reader to piece together the materials & methods would be very helpful. Fourth, both chapters "Biogenic opal" and "Alkenone analysis" do not clarify what core material was used. It is unclear what core was used for what analysis. Similarly, it is unclear what cores were used for the spectral analysis. U1486 is referenced in the text, but the spectral analyses do not appear in Figure 3. Fifth, it appears the authors stacked a suite of records in Fig. 1b. However, the description of this stacking process is very limited. Questions remain regarding why was 500 years chosen as the interpolation time step? What exactly does "time dependent mean" refer to? I suggest the authors include a mathematical formula in the supplement. How exactly was the error envelope plotted in Fig 1b calculated? What errors are included in the error margins? Does it include age model errors? Several stacking approaches are published, did the authors choose their own approach or used an already published approach?

(b) I would like to see a more careful approach regarding age models. First, the "age model" section in the methods clarifies that core MD01-2380 was tuned to U1486 using planktic foraminifera. This is a somewhat unusual method, as usually methods call for benthic foraminifera to be tuned to each other and ultimately to the orbital parameters. Maybe the authors can include a sentence or two on what age model errors are expected using this approach. Second, it is currently unclear whether the foraminifera d¹⁸O record of U1486 is planktic or benthic. I assume planktic? It is important to know whether the age model compares "apples with apples". Third, there is no description of how the age model for U1486 is constructed, or any of the other cores. What about the age models for the other cores listed above in chapter "Isotope measurements"? Fourth, how did the authors deal with age model differences when stacking the records in Fig 1b? How did they incorporate age model errors in the error envelope?

(c) The discussion on ITF mixing ratios, as well as the use of the term "ITF", vs. "Banda Sea" is confusing. I would like clarity

for the reader in terms of what the authors mean by "ITF". I suggest the authors move away from writing about the "ITF" and focus on their study site which is in the Banda Sea. The mixing of water masses is currently discussed in the introduction as well as the methodology section as "Text S1, S2" etc. I find this structure very unintuitive, and not supportive for the manuscript. I suggest that the authors take the information out of the "Texts" and include them in the manuscript. Kindly describe the water masses and their percentages of total water as they appear in the Banda Sea, and then in the same fashion as they appear in the east Indian Ocean, where they can be called the "final" ITF. This should include sources that may not channel via the Banda Sea. I suggest that the authors then do a back-of-the-envelope quantification of the fraction of southern-sourced waters in the Banda Sea and the "final" ITF in terms of % of modern volume flow. Then this quantification can be taken back in time. Like this, the reader will be able to follow how much the contribution impacts the records at the Banda Sea core site, and subsequently the "final" ITF.

(d) After having noted my own comments, I read the rebuttal and noticed that many of my comments already appeared in the previous round of reviews. For example, previous reviewers already asked for a more detailed method section/supplement, and a more nuanced discussion on the ITF vs the Banda Sea. I would like to encourage the authors to respond to reviewer comments with more enthusiasm and improve their manuscript. I also believe some comments have not been addressed. First, the request for a more in-depth discussion of the spectral analysis results. I encourage the authors to satisfactorily discuss what implications their results have on larger scale changes, such as sea level variability. The authors state that any interpretation would be "purely speculative", but I respectfully disagree. In my view it is quite an interesting finding that the authors don't see a strong 100kyr cycle given the sea level dynamics in the region over the Pleistocene (e.g. Nuber et al., 2023). What hinders this signal to appear in the Banda Sea? Second, I do not follow the authors' response on the C/N ratio. What does the attached plot show? What samples are plotted here? How exactly are the authors justifying their method? I suggest including this plot and this method validation into the supplement and providing much more detail. Third, the authors claim in the rebuttal that the $\delta^{15}\text{N}$ in the Banda Sea is not depth dependent, only as distance to land. Surely degree of stratification would influence the $\delta^{15}\text{N}$ as different water masses enter the Banda Sea? Is that not one of the concepts that the authors use to calculate the amount of southern-sourced water?

Line by line comments

Main text:

Line 83: Kindly include a more in-depth discussion on the endmembers. What are the endmember $\delta^{15}\text{N}$ values for these different water masses? Are they stable through time? Are they distinct water masses in the Banda Sea? Or vertically mixed already? A more nuanced discussion of the ITF is needed.

Line 107: the sentence states "Offset between southern hemisphere record averages 2.3permil... Figure 2". I am slightly confused, as the previous sentence in line 105 already quotes 2.3permil for the difference in the northern WEP stack. In Fig 1, the southern hemisphere stack is clearly less offset than the NH stack. In Fig 2, why do the authors calculate a $\delta^{15}\text{N}$? Presumably the authors want to reference Fig 2c, rather than the entire Fig 2? Are the authors referring to the average value of 2c? If so, it would make sense to briefly explain why this matters, what it expresses, and include a line to show where the average is.

Line 113: Can the authors reference this sentence. What modern observations are we talking about? Are they plotted anywhere?

Line 127: Not sure what the authors mean by "absolute minimum estimate"

Line 138: These claims are hard to see in the provided plots. Maybe include shaded bands to highlight precession or obliquity pacing for the eye. Or include cross-plots of orbital parameters and $\delta^{15}\text{N}$ to show these correlations visually.

Line 146: how do Figures 3, S3 support this statement? Fig. 3 only shows the spectral analysis of core MD01-2380, and Figure S3 does not show any spectral analysis. Is there another figure that might be leveraged to show this more clearly?

Line 159: It appears that there is a water mass here which provides a substantial amount of water for the ITF, but might be left out in the authors' proxy, due to the lack of nutrients. Have the authors considered this when calculating their southern-sourced water contribution? As stated above, I recommend the authors do a calculation where they clearly state what the southern-sourced water contribution is in Sv, and where they account for all other sources in Sv too.

Line 186: I am unsure how Figure 3 supports this statement. Figure 3 does not show spectral analyses from U1486.

Line 194: I do not see that the two records are particularly similar across the entire record. I am surprised by their differences. That said, these are all "subjective by eye" statements. I suggest the authors use some statistics to explore how similar/different these records are. For example, the authors can use a sliding window and compute a correlation coefficient. Or calculate differences to highlight regions of similarity vs difference. I am encouraging the authors to explore their own methods depending on what it is they want to show.

Line 198: I have difficulty seeing the correlation between orbital parameters and the records. I suggest re-plotting in a different way (ie using cross-plots or other methods as suggested above).

Line 200: reference a plot, where do we see that peak primary production occurs during austral winter?

Line 205: reference a figure, Where do we see that obliquity is only evident in biogenic opal not alkenones? What core are we talking about?

Line 210: define "substantial". I am looking forward to reading a number in Sv or as % of total Sv here.

Reviewer #5

(Remarks to the Author)

Review of "Significant Southern Hemisphere contribution to the Indonesian Throughflow over the last 800,000 years" by Kienast et al.

A few pages into this paper I realized that I should perhaps not have agreed to review it. Reviewers 1-3 clearly have more expertise in tracers and in Pacific Ocean water masses than I do. But this makes me more of a general reader, and since

articles from this journal are intended for a broad audience, I might be able to contribute something.

I confess to being initially surprised that the conclusion that a SH source for the Throughflow should be novel: as Reviewer 1 points out, Godfrey's island rule predicts a net northward flow east of Australia of about 15 Sv.. This transport is the net Sverdrup transport minus the transport of the western boundary current along Australia's east coast. So really ALL the ITF Throughflow has a SH origin, and the debate seems to be the partitioning between waters that take a more or less direct route into the Indonesian Archipelago and those that have been seen substantially modified as a result of residence in the NH. The authors attempt to quantify SH influence over the past 800,000 years by comparing the history of ^{15}N from a sediment core taken in the central Indonesian Archipelago to cores taken from locations of established NH or NW origin. Reviewers 1 and 3 are favorable, and all three reviewers acknowledge the importance of the data. Review 2 is unhappy with the original submission, feeling that the novelty of the findings were overstated, that insufficient coverage was given to recent literature, and that elements of presentation was poorly framed. My reading of the revised version suggests that sufficient caution is exercised in stating conclusions, and that recent literature is at least widely cited. I do think that despite the improvements the paper is still going to be challenging for an outsider like myself, and I also have a few questions about the analysis, but my impression is that the work is nearly publishable.

1) It is well known that the Lombok and other passages that drain the Indonesian Archipelago undergo frequent flow reversals as a result of Kelvin waves of Indian Ocean origin. Murray and Arief (Nature, 1988, Fig. 3) show examples. Whether the resulting northward inflows of water of Indian Ocean origin make their way into the Banda Sea, I do not know. Is this a consideration?

2) Is anything known about changes in Pacific surface wind field over the past 800,000 years? This could potentially effect Godfrey's estimate of the net northward transport.

3) I don't agree with your response to Reviewer 2's questions about G-IG sea level variation. It's a very natural question the average reader will have since sea level changes could have interesting effects on the way that the ITF is routed. And, if the ITF is really "choked", this may have upstream ramifications. I don't think it would "confuse the reader" to mention this, perhaps as just a caveat. And perhaps significant to point out that you do not see evidence in the core samples of such variability (if I understand that correctly).

Other points:

Line 58. Surface temperature is certainly not conservative. Neither are nutrients.

Line 78 refers to site MD01-2380 as labeled by a star in Fig. 1a, but the figure has the star labeled as MD80.

Line 109. If you are going to refer to the U1486 curve in Fig. 1b. it would be helpful to label it.

Line 119. You use only U1486 to represent $\delta_{15\text{N}}^{\text{South}}$, apparently because you feel it is best correlated with MD01-2380 in Fig. 1b. MD20 is geographically quite close to U1486 but is not used, which raises some questions in my mind. This also relates to what is written later on lines 125-127. I suppose that, just for comparison, you could have done a second calculation using the average of the SH sources to represent $^{15}\text{N}_{\text{South}}$.

Version 2:

Reviewer comments:

Reviewer #4

(Remarks to the Author)

Manuscript Review for

Significant Southern Hemisphere contribution to the Indonesian Throughflow over the last 800,000 years

Overall remarks

As previously stated, Kienast et al. provide an exciting study that now clearly disentangles the southern from northern nutrient sources to the ITF using nitrogen isotope tracers. I congratulate the authors on their new manuscript that now answers all my comments and, in my opinion, reads logically to a wide audience. The results and hypotheses are clearly stated, and the data is logically presented. I am pleased to see the fleshed-out version of the methods, which is now also much easier accessible. Some very minor comments below, but ultimately, I am happy to see this manuscript published in its current form.

Warm regards,
Sophie Nuber

Minor comments (lines refer to the clean version):

Line 71: The authors reference Fig. 2a, I believe they want to reference the MD01-2380 record which would be Fig 2b?

Line 164: This hypothesis is very compelling, however it left me thinking: how can the authors disentangle north from south, if both are ultimately driven by the same southern signal? The authors describe well that the northern Hemisphere WEP "reflects" (line 162 – 164). However, it may be beneficial to have one sentence here clearly stating what the southern signal reflects. I presume it reflects only the nutrient-rich southern signal, without the overprints experienced by the north?

Line 287: At risk of being pedantic, it would be useful for an interested reader (like myself) to have a brief statement that the age model of U1486 is based on alignment of benthic d^{18}O to LR04.

Line 324: It would be beneficial to see the modern values included in Figure 1 as symbols for both stacks.

Figure S5: The figure attached separately to the upload system seems to have cut-off the precession axis. From Figure 2 I assume that precession is plotted with an inverted y-axis? I suggest making a short note of the inverted axis in the figure caption.

Line 363: I noticed the additional line in the Acknowledgements. I send the authors my condolences and look forward to seeing this manuscript published.

Reviewer #5

(Remarks to the Author)

The authors have addressed my comments to my satisfaction. I am ok with publication of the article. Larry Pratt

Reviewer #1 (Remarks to the Author):

In this paper, Kienast et al. use the $\delta^{15}\text{N}$ of the bulk organic matter in eight sediment cores to estimate 1) the relative amounts of Northern and Southern Hemisphere water in the Indonesian Throughflow (ITF) and 2) the way that the sources vary over time. Kienast et al. find that Southern Hemisphere water makes up more than half the ITF most of the time. This is a very interesting and compelling result. Given the longstanding controversy regarding the sources of the ITF, it is somewhat surprising that this simple analysis was not done years ago.

The analysis in the paper is very straightforward, and I am inclined to say that the paper could be published almost as is. An important clarification is needed, however. The ITF is not a single entity. It is spread over 1500 m or so and the water near the top is quite different from the water near the bottom. The part of the ITF that is examined in this paper is the near-surface and upper thermocline parts of the flow. This is acknowledged in the paper but the consequences are not.

The basis for a Southern Hemisphere source for the ITF was laid out by Stuart Godfrey in 1989. Basically, the westerly winds blowing along the southern boundary of the Pacific basin at ~45 deg. S push a large volume of water northward into the basin. If there is a gap along the western boundary of the basin, as there is in the Indonesian Seas, the water that is pushed across 45 deg. S exits through the gap. Godfrey predicted that the flow through the gap should be about 15 Sv -- an astonishingly good prediction.

The water that actually flows through the Indonesian Seas, however, seems to have a northern source, as had been pointed out a few years earlier by Arnold Gordon in 1986. Gordon thought that a fairly large volume of old NADW upwells in the North Pacific and returns to the North Atlantic via the ITF and Agulhas eddies. Gordon saw the Throughflow as part of the overturning of NADW. These two ideas about the Throughflow couldn't be more different.

Kienast et al.'s results, taken at face value, tilt toward the Godfrey's approach. The problem is that Godfrey, in his original paper, was not the least bit concerned with the properties of the water that flows into the Pacific across 45 deg. S. Kienast et al. need to be concerned about these properties because the properties of the water that flows across 45 deg. S are very different from the properties that are monitored by their sediment cores in the western Pacific.

The water that enters the Pacific via the winds along 45-50 deg. S is most readily characterized as Subantarctic Mode Water (SAMW). SAMW is roughly a 50:50 mix of old deep water that is drawn up to the surface south of the ACC and subtropical thermocline water from the South Pacific. The old deep water in SAMW is a signal that the ITF is a hybrid flow, some of the ITF is wind-driven, some is part of the ocean's overturning. Most importantly, SAMW is quite cool (because of where it forms). Kienast et al.'s near-surface upper thermocline southern source, in contrast, is quite warm. So, if Kienast et al. are correct, a major water mass transformation must be taking place along the path that SAMW takes from 45 deg. S to the Indonesian Sea.

I published a pair of papers in 2019 (with co-authors Ellen Druffell, Robert Key, and Eric Galbraith) that examined this transformation (Toggweiler et al., Upwelling in the Ocean Basins North of the ACC, Parts 1 and 2, JGR-Oceans, 10.1029/2018JC014794 and 10.1029/2018JC014795). We used the surface D14C to monitor the circulation. D14C is similar to d15N in the sense that both tracers are weakly modified by contact with the atmosphere. Our results showed that much of the southern water that enters the Pacific across 45 deg. S is upwelled to the surface off Peru. We were also able to assess the surface waters that enter the Indonesian Seas, and we found that the southern water from Peru is the main component. Our results are thus quite consistent with the results in the present paper. The latter finding is discussed on page 14 in the Supporting Information for Part 1.

In Part 2, we simulated the surface distribution of D14C in a modern climate model and found that the none of the subantarctic water from 45 deg. S in the model was reaching the surface off Peru. It tended to exit the Pacific as a cool subsurface flow instead. Thus, the inflow across 45 deg. S in the model remained unmodified as it made its way to the Indonesian Seas. This discrepancy is discussed on pages 10-13 in Part 2. The discrepancy then led us to a hypothesis to explain how so much SAMW manages to upwell off Peru in the real world.

Our hypothesis is that the formation of NADW in the North Atlantic draws away the warm buoyant water that is piled up in the western Pacific by the easterly winds in the equatorial zone. The volume of warm buoyant water that is drawn away in the west thereby determines the volume of cool water that is upwelled to the surface off Peru. Our model did not manage to draw any of the piled up water in the west away, so there was much less upwelling off Peru in the model than observed. As a result, the inflow across 45 deg. S in the model ends up leaving the Pacific as a cool subsurface flow.

The reason that I mention all this is that our model ended up generating a circulation that is somewhat like the one that Godfrey described, i.e., the water that leaves the Pacific via the ITF is largely the same water that enters the Pacific across 45 deg. S. If the real ocean worked this way, Kienast et al. would not see any evidence for a southern source in their sediment cores. (The southern outflow would be too deep to influence the organisms that produce the organic matter that is extracted from the sediments.)

So, IMHO, Kienast et al. are able to "see" the southern source in their cores because the inflow across 45 deg. S is transformed off Peru. The paper would be greatly improved if it included this perspective.

J. R. Toggweiler

We thank J. R. Toggweiler for his valuable feedback and detailed explanation. This important mechanism is now added to the revised version of the discussion (lines 118-153, and particularly in lines 131-153). **Please note that line numbers refer to the clean version of the revised manuscript.**

Reviewer #2 (Remarks to the Author):

General Written Evaluation

First of all, I want to apologise for taking longer than expected to complete this review. While dangling the potential of a significant scientific paradigm shift in front of the reader, this manuscript woefully falls short in transporting this message in its current structure. It was thus difficult to review, as it presents excellent data but wraps it in a somewhat problematic narrative that often fails to establish key concepts early enough or downright misrepresents large portions of a large body of especially more recent scientific literature dealing with the same study area. The manuscript titled 'Significant Southern Hemisphere contribution to the Indonesian Throughflow over the last 800,000 years' tries (but has failed to convince me) of the fact that previously a critical part of the Indonesian Throughflow water mass contribution has been completely ignored in previous literature. Namely, they tout the hypothesis that the southern hemisphere water contribution to ITF waters is completely ignored in the literature and that their dataset is the first to allow proper evaluation of said oversight. Here, I wish to emphasise that their data shows compelling evidence for the southern hemisphere water influx if taken at face value. I simply disagree with how they tried to discredit the rest of the scientific community to generate an overinflated impact of their data – and I know this is probably done primarily to fit the "flashy" style of many articles that are published in "high-impact" journals such as Nature Communication. A trend, I feel, is wholly unnecessary if one can let compelling data speak for itself.

With that (admittedly very harsh) opening statement, I wish to take a short step back and explain why I have arrived at this overall negative conclusion after several re-reads of the text and despite a strong motivation to like this manuscript.

Despite all the negativity, I still think the author's data is valid and valuable. But – and this is crucial – the authors did not manage to be convincing in framing this data and connecting it to the overall significance of the region. Even more problematically, they also did not do their due diligence in performing even rudimentary tests to ascertain their proxy assumptions are valid – they cite data and rattle off reasons the reader is supposed to believe (disclaimer: I do 'believe' them, but only by the benefit of being very somewhat familiar with the data basis and the literature). I, therefore, wish to fully emphasise that I do not make this statement lightly, especially considering that many of the authors are well-regarded experts in the field and have even (co-)authored several papers dealing with such proxy validation. I am baffled by the lack of due diligence regarding data evaluation.

The fact that the authors do not even bother to include such fundamental principles in their methodology or their supplements is not adequate for any reputable scientific journal, let alone a journal intended for a general scientific audience such as Nature Communications (although they do touch upon it in the discussion chapter). I thus feel that this manuscript needs a lot of additional work on the drawing board before it can be considered for publication in a scientific journal. If the authors wish to publish it in a journal geared towards a general scientific

readership (such as Nature Communications), they will also need to critically re-evaluate some of their visual style choices, as well as restructure their frame story and writing clarity. I found their frame story lacklustre in its impact, the beginning (not necessarily the latter half) of their data discussion distracting, and their figures confusing and challenging to read.

Below, I have illustrated the basis of this conclusion through a set of general chapter-specific comments and specific line-by-line comments throughout the manuscript. I am sorry I cannot be more positive in my assessment of this work, but the overall presentation of the data does not live up to the impact the authors wish it to have—or that the data itself would deserve.

We thank the referee for finding our *'excellent data'* *'valid and valuable'* that show *'compelling evidence'*, and for the *'believe'* in our reasoning. These are the most important missions of a scientific paper. Regarding the overall criticism, we find the evaluation of the referee in this part too general and not specific enough to be addressed in a point by point fashion. Complains such as *'I simply disagree with how they tried to discredit the rest of the scientific community'*, or *'I am baffled by the lack of due diligence regarding data evaluation'*, or *'they will also need to critically re-evaluate some of their visual style choices, as well as restructure their frame story and writing clarity'* won't help us improving the manuscript. Some of these criticisms are mentioned in the chapter-specific comments below, and addressed in our point-by-point response. **Please note that line numbers refer to the clean version of the revised manuscript.**

Chapter-specific Comments

Abstract

The abstract is scientifically imprecise and overemphasises a narrow viewpoint of an unknown to inflate the study's significance. Such imprecision should be remedied, as ample work on the region's modern oceanography adequately describes the ITF system's complexity.

The abstract has been revised accordingly to better reflect the current state of knowledge on the ITF, with a focus on the knowledge gaps in past ITF variations. In light of this, we respectfully disagree with the referee's statement that "ample work on the region's modern oceanography adequately describes the ITF system's complexity". While we acknowledge the contributions of measurement campaigns such as INSTANT and IMOS, these efforts were limited by their duration and scope, and insufficient to fully capture the contributions of source waters to the ITF. We refer to the review article by Sprintall et al. (2019, doi:10.3389/fmars.2019.00257), authored by 28 physical oceanographers, several of whom have been studying the ITF for over three decades, which underscores the persistent gaps in our understanding. There is a five-page discussion of shortcomings and a to-do list and the end, pointing out that "there remains a scarcity of temperature, salinity, and biogeochemical profiles from within the Indonesian seas. This precludes building even a seasonal picture of water mass variability, let alone tracking changes from year to year". We hope this clarifies our position.

Introduction

I am somewhat worried about how the authors push an agenda and narrative within this introduction that often critically omits the nuance of previous works. This generates a (perceived) problem in the literature that the authors now set out to solve. This, I feel, detracts from the actual importance and validity of the author's dataset (albeit some methodological misgivings I may have with it; see below)

We believe that the Introduction section of a scientific paper should be concise and focused. Rather than offering an exhaustive review or delving into the nuances of the existing literature, it should aim to provide readers with an overview of the key knowns and unknowns that are directly relevant to the specific study at hand. Our introduction consists of 1st paragraph introducing the modern ITF, 2nd paragraph on the source waters of the modern ITF, 3rd paragraph on the reconstructions of the ITF, 4th and 5th paragraphs introducing our method and its applicability, and the last paragraph outlining the paper.

We think that the referee's concern is mainly about the 2nd and 3rd paragraphs, which are now revised (lines 23-44). See also our line-by-line responses to the referee's comment on the introduction section below.

Results and Discussion

Overall, the data presentation is lacklustre. Figure 1 is not labelled correctly. The erratic style choices in how data is visually represented make this manuscript completely unreadable, especially for people with colour-impaired vision. The authors will need to fix both the clarity of their discussion and the clarity of data visualisation before any (more critical) discussion of their data in terms of scientific interpretation can be started.

We have changed the color scheme and labeling of all figures in the revised version. The discussion section is modified as well (see the line-specific response below).

Furthermore, considering that this study spans the last 800 kyr, I would have appreciated at least some discussion on the effect of sea level on varying ITF connectivity. The Macassar straight, especially, is well known to have had significantly altered connectivity during the late Pleistocene glacials. The authors ascribe an "over-emphasis" of the Macassar Straights' contribution to the story but fail to account for sea level-driven ITF changes entirely. This is a critical problem that needs to be addressed, especially since they do cite some Plio-Pleistocene studies that touch upon these changes in the context of ITF dynamics (e.g. their ref18).

Since the 100-k G-IG cyclicity is not observed, as also noted by the referee (see below), we refrain from addressing this aspect in the original submission. We believe that discussing G-IG variability not captured by our data would be purely speculative and could detract from the main narrative, potentially confusing the readers.

Finally, the authors offer a very well-crafted discussion of their data and manage to place it into a large global earth system context in the latter half of their results and discussion chapter. However, this discussion makes it even more difficult to accept the statement made in their introduction, as they offer numerous citations to again prove that the connectivity and contribution of southern hemisphere intermediate waters in the tropics is well known – I cannot get over the fact that the authors decided to eschew the (undeniable!) importance and impact of their data in such a way, that it partly and erroneously represents the large and growing body of literature dealing with the Pleistocene to recent dynamics of the Indonesian Archipelago. I am very disappointed that the authors discuss precession and obliquity patterns but fail to contextualise these changes in terms of large-scale glacial-interglacial variability – especially sea level-related restriction of large portions of shallow pater ITF connectivity. This would better support their (not necessarily new, but certainly very well supported by data) ideas regarding the Indonesian Archipelago's role as a key "choke point" in the global thermohaline overturning circulation and thus also nutrient fluxes in the ocean.

Please refer to our previous and following comments. The referee's concerns pertain either to the Introduction section or to the sea-level and orbital variability, all of which have been addressed above or below.

I am also disappointed that the authors fail to emphasise the importance of direct pathways of AAIW/SAMW into the lower latitudes and the northern hemisphere and how these intermediate water pathways directly influence upwelling in the open ocean compared to their data from the enclosed Banda Sea. – that being said, I am well aware of the lack of congruency between their data and G-IG forcing, both in the spectral and the visual domains. That, however, does not mean the authors can skip discussing it.

The pathways have been addressed in the revised version, in accordance with the first referee's suggestion (lines 131-153). Regarding G-IG variability, see our response above. We believe that discussing G-IG variability not captured by our data would be purely speculative and could detract from the main narrative, potentially confusing the readers.

In conclusion, I do not feel this manuscript is ready for publication despite the authors' undeniable data quality and considerable effort in data synthesis. The text, figures, and discussion are simply not on a level that reflects both minimal quality standards and the state-of-the-art of the current literature. This is especially problematic, as even the authors' starting premise does not reflect the general understanding of modern-day oceanography of physical and chemical oceanographers or the paleoclimate community (based on my experiences).

This appears to be a reiteration of the referee's earlier points, which seem to focus on the perceived absence of certain aspects or studies in the introduction. It also suggests that modern-day oceanography is well understood, a position we respectfully disagree with, given the ongoing uncertainties in the field (see e.g. Sprintall et al., 2019).

Sprintall, J., Gordon, A. L., Wijffels, S. E., Feng, M., Hu, S., Koch-Larrouy, A., . . . Setiawan, A. (2019). Detecting Change in the Indonesian Seas. *Frontiers in Marine Science*, 6(257). doi:10.3389/fmars.2019.00257

Methods

I am very concerned that the authors do not evaluate the source of their organic matter further. Generally applying the TOC:TN ratio and $\delta^{13}\text{C}_{\text{org}}$ should have been measured and evaluated to show the marine origin of the bulk OM they measured. Please follow the recommendations of Meyers (1994) here:

Meyers, P. A. (1994). Preservation of elemental and isotopic source identification of sedimentary organic matter. *Chemical Geology*, 114(3–4), 289–302. [https://doi.org/10.1016/0009-2541\(94\)90059-0](https://doi.org/10.1016/0009-2541(94)90059-0)

Before the authors ascertain the marine origin of the presented bulk organic matter $\delta^{15}\text{N}$ record, I cannot allow this work to be published in good conscience. Again, I do not doubt the validity of the data; I am just baffled by the authors' lack of due diligence in terms of method justification here.

We present three lines of evidence to demonstrate that the $\delta^{15}\text{N}$ values of MD01-2380, as well as the other cores included in our study, are not influenced by terrigenous input.

First, the C/N ratio (average = 8.6) and $\delta^{13}\text{C}_{\text{org}}$ (average = -20.9) measured on multiple samples throughout the core clearly indicate that the bulk organic matter is of marine origin (see figure below).

Second, Core U1486, retrieved near the Sepik River mouth off the coast of New Guinea, should theoretically show the strongest influence of terrigenous material. However, as discussed by Lambert et al. (2022), the TOC/TN ratios and $\delta^{13}\text{C}_{\text{org}}$ data for U1486 suggest that the Sepik River has had minimal or no impact on the organic fraction of the core. Moreover, the authors observed no systematic shifts in $\delta^{13}\text{C}_{\text{org}}$ or TOC/TN ratios between glacial and interglacial periods, further supporting the idea that sea-level driven terrigenous influence does not affect $\delta^{15}\text{N}_{\text{bulk}}$ variability (Lambert et al., 2022).

Finally, the $\delta^{15}\text{N}$ signal in the MD81 and MD67 records is consistent despite these cores being retrieved from vastly different settings. Given the higher sedimentation rates at MD81, we would expect a stronger influence of terrigenous material. This regional consistency in $\delta^{15}\text{N}$ values across different cores—despite their distinct environments and sedimentation rates—reinforces the conclusion that $\delta^{15}\text{N}$ is independent of terrigenous input. This information has also been added to the SI (Text S1).

Supplements

I am putting this subchapter here in the hopes that the authors, for their next attempt at submission, at least will consider adding a much more extensive supplementary section that can better justify their arguments. While short-form papers are excellent at succinctly conveying compelling science stories/news, the meat, data/interpretative justification, and extensive hypotheses testing should always be present somewhere – usually in the supplements. Considering that the authors purport to present such a paradigm shift in this manuscript, I am consequently missing a systematic and extensive discussion of these concepts in a broader context. Hence, why are there no supplements except four figures, which could each have easily been integrated as part of one of the main figures? Fig. S1 could be part of Fig. 1; Fig. S2 could be part of Fig. 2, and so on...

The Figures and the Supplementary Information have been changed in the revised version.

Line-specific Comments

Abstract

Line 3: Please do not use iconographic in this context. It has a narrowly defined meaning, and although it can be used in this context, it may be not very clear to general readers.

Done

Line 6-8: I wholeheartedly disagree with that statement. Major sources of subsurface (lower thermocline) water masses flowing through the ITF have always been attributed to the Pacific South Equatorial Current. Everybody with a basic understanding of physical oceanography will grasp that concept. This amount of imprecision is unacceptable in a scientific abstract. To quote from

In the revised version, we have clarified this by stating: little is known about the hemispheric origin of the water masses contributing to its overall transport in the past (lines 5-6).

Introduction:

Line 26: I am somewhat worried the authors may not have fully grasped the information they referenced [6]. It has always been known that southern hemisphere waters contribute appreciably to the ITF that enters Halmahera (especially in the geologic past!). A more detailed reading of the literature will also reveal that the most significant part of the southern hemisphere sources water flow at lower thermocline and sub-thermocline depths.

Here are some suggestions for further reading:

Gordon, A. L., & McClean, J. L. (1999). Thermohaline Stratification of the Indonesian Seas: Model and Observations*. *Journal of Physical Oceanography*, 29(2), 198–

216. [https://doi.org/10.1175/1520-0485\(1999\)029<0198:tsotis>2.0.co;2](https://doi.org/10.1175/1520-0485(1999)029<0198:tsotis>2.0.co;2)

Tillinger, D. (2011). *Physical oceanography of the present-day Indonesian Throughflow*.

Geological Society, London, Special Publications, 355(1), 267–

281. <https://doi.org/10.1144/sp355.13>

Feng, M., Zhang, N., Liu, Q., & Wijffels, S. (2018). The Indonesian throughflow, its variability and centennial change. *Geoscience Letters*, 5(1), 3. <https://doi.org/10.1186/s40562-018-0102-2>

We appreciate the reference to additional sources, which we have been aware of before writing the original draft. We note that all three sources support our basic premise (and the message of this sentence) that according to the literature, the majority of the ITF is of northern Pacific origin, and that the contribution of the Southern Hemisphere is less than 20%. Here are additional sources with similar estimates of the South Pacific contribution:

Sprintall et al. (2014; doi:10.1038/ngeo2188): South Pacific contribution estimated at 10–20%

Liang et al. (2019; doi:10.1029/2018JC014926): South Pacific contribution ~18%

Gordon (2005; doi:10.5670/oceanog.2005.01): South Pacific contribution is roughly 10–20%

Line 34 – 41: This is a gross oversimplification of the literature cited and a limited view of the large body of works available. I do not feel that this is how a scientific argument should be built (irrespective of whether I agree with its validity, I most certainly do not). Regardless of my misgivings in how this received issue is framed, I am most baffled by how the authors can cite citations 15-18 in support of such an argument. Finally, I find the term 'tacitly' somewhat inflammatory here. It seems to be placed with the sole intention of verbally enforcing a scientific gap that, in all actuality, does not even exist. Furthermore, it becomes clear already by this portion of the manuscript that some key aspects in the behaviour and mixing of ITF water masses in the Indonesian Archipelago may not have been considered fully by the authors.

The referee is certainly not expecting a comprehensive review of the existing literature. In response, we have rephrased this section of the introduction to highlight the knowledge gap, by focusing on the most essential background rather than attempting an exhaustive review of the topic (lines 34-44).

Line 41 – 45: I agree with that statement. However, I am not happy with the author's attempt to bend the evidence of physical oceanographic observations to fit their narrative, which is pervasive in their introduction.

The statement here is that the error of proxy-based reconstruction of past temperature and salinity is too large to decipher between the different sources of the ITF. This is a fact as the error of these methods are known and too large. We clarified this in the revised version (lines 41-44).

Line 46 – 52: Yes, that is true, but the authors do not measure nitrate in their study. They measure bulk sedimentary organic matter nitrogen isotope ratios. Therefore, local denitrification and nitrogen recycling of the sediment, as well as riverine influx of terrestrial nitrogen sources such as land plant-derived organic matter from the Indonesian Archipelago, will need to be considered in more detail to substantiate proxy claims the authors put forward in this manuscript.

The referee is correct in pointing out that we do not measure nitrate but bulk sedimentary $\delta^{15}\text{N}$. We therefore state in two paragraphs that ‘bulk sedimentary $\delta^{15}\text{N}$ faithfully records past variations in subsurface/upper thermocline nitrate $\delta^{15}\text{N}$ in the WEP’ (lines 52-57) and especially in the Banda Sea (lines 68-74). Water column denitrification does not occur in the Banda Sea because of the well-oxygenated water column. In addition, the Banda Sea core is too deep for nitrogen recycling of the sediment. Regarding the possible input of terrestrial nitrogen, please see our response above. All this information is added to the revised version (Text S1).

Line 53 – 58: All except one (Citation 26), all citations supporting this argument have been authored or co-authored by one or more of the authors of this manuscript. While I do not doubt the validity of their previous studies, it may indeed explain some of my earlier misgivings concerning the tone of the introduction so far.

The nitrogen isotope community is rather small compared to other fields of paleoceanography or biogeosciences. We thus cannot be accused for being involved in many studies on equatorial Pacific nitrogen isotopes, including working group synthesis publications with 40 authors (ref 28). To the best of our knowledge, we have cited here all the available studies from the WEP that presented $\delta^{15}\text{N}$ in both bulk sediments AND co-located water column nitrate, leading to our statement that ‘bulk sedimentary $\delta^{15}\text{N}$ faithfully records past variations in subsurface/upper thermocline nitrate $\delta^{15}\text{N}$ in the WEP’.

Results and Discussion

Line 68 – 88: I have read these paragraphs multiple times, and despite being extensively familiar with the study region in terms of modern oceanography, IODP, and MD site locations, this was more than just hard to parse. The conclusion here is thus: A more general reader simply interested in the core message of this work will be lost before any meaningful discussion has commenced. Please revise for clarity. This includes better visualisation of the site location (colour difference is not enough!)

We have completely revised the colour scheme of the figures and rephrased this part to improve readability (lines 68-84).

Line 73 – 76: Yes, very curious. I would love to see more discussion on this fact. Possibly focused on (discounting) the possible contribution of terrestrial organic matter $\delta^{15}\text{N}$ from the not insignificant riverine influx in the Indonesian Archipelago and also north in the South China Sea, including the Mekong and Yellow Rivers.

These lines explain why $\delta^{15}\text{N}$ in the Banda Sea is isotopically *enriched*, i.e., *heavier*. Any possible contribution of terrestrial organic matter would decrease bulk sediment $\delta^{15}\text{N}$, not increase it.

Line 89-101: This paragraph finally, for the first time in the manuscript, provides a neutral and easy-to-follow discussion on the different sources and their potential contributions. However, as stated above, I am still missing a text of the null hypothesis to disregard the local influx of organic matter into the Banda Sea (disclaimer: I do not doubt that the data and interpretation of the authors will most likely remain valid, but it still needs to be done in the interest of due diligence of data analyses).

See the above response. The $\delta^{15}\text{N}$ values are enriched in the Banda Sea, excluding a potential terrestrial contribution to bulk sediment $\delta^{15}\text{N}$.

Line 116 – 117: This is a pretty offhanded sentence. The authors should at least reference (or better yet include) such a mass balance model if they mention it!

We apologize for the confusion here. In the original version, we used ‘simplified isotope mass balance’ and ‘mass balance model’ interchangeably. This has now been changed to ‘calculated’ (line 107).

Line 126 – 131: I am not considering precessional-driven climate changes in the tropics. The authors immediately move to water mass stratification but ignore the intertropical convergence

zone and the interlinked monsoonal changes in this discussion. Again, they need to confidently discount this to substantiate the presented hypotheses.

We use $\delta^{15}\text{N}$ as a water mass tracer in this study. While both the Hadley circulation and ITCZ are important for discussions on tropical climate, these topics would shift the focus of our study, which is primarily centered on ocean circulation. We have modified the discussion here, in line with the suggestion by referee #1 (lines 118-153).

Line 140 – 148: I don't doubt precessional control over equatorial upwelling in the Pacific. But I am beginning to doubt that the authors have considered all the nuances underpinning this system, especially in the context of seasonally shifting Hadley circulation. Also, in the end, they mention ODP Site 1012. Would it be possible to show this on a map and, ideally, graphically illustrate the connection to the discussed dataset? This would help readers understand the argument the authors are trying to make here.

The first part of the referee's comment is addressed above. A new figure illustrating the connection and core location has now been added to the SI (Fig. S3).

Line 146: EUC has never been defined. Please do so.

Done (line 148)

Line 148 – 156: This brings back the discussion nicely, but I am still a bit confounded that the authors show the close comparison in ODP Site 1012 and Site MD06-3067 but fail to display this connectivity accurately in Figure 1. This detracts from the quite powerful statement they make here.

A new figure illustrating the connection and core location has now been added to the SI (Fig. S3).

Line 159 – 167: I have to commend the authors here. This is a well-crafted explanation and reasoning. It reflects the state of the art of our understanding and presents how their data fits into it. However, I still fail to see how their data constitutes a paradigm shift on the level they hint at in their introduction.

Thank you. The connection between the equatorial and southern Pacific has been shown for deglaciations, while we infer a continuous and stable connection over the past 800 kyrs. We hope that our revised introduction now provides a better context for this section, which seems to bother the referee the most.

Line 170 – 184: First: This would be better suited in the methodology and/or earlier in the discussion stage. Second, I feel this is an inadequate way of assessing data validity, as it hinges on the "because we say so" approach. No concrete testing is applied.

In response to the referee's comment, we have moved this section to the SI (Text S1), complemented by a discussion on possible terrestrial contribution. Far from 'because we say so', we refer to (ref 70) and point the reader to the results presented in this study.

Line 197-204: Yes, but that is also true for any other upwelling region globally, so please be more specific on how it supports their line of argument regarding the uniqueness of their $\delta^{15}\text{N}$ data. Biogenic opal essentially shows the same after all.

The close agreement between bulk sediment $\delta^{15}\text{N}$ and opal provides an entirely independent line of evidence, and the obliquity power observed here is indeed a critical piece of corroborating evidence for our interpretation of the $\delta^{15}\text{N}$ record.

Line 209 – 214: This is very weakly developed regarding scientific arguments. The authors ignore that the Indian Ocean has a direct (and I would argue very well understood) connectivity to the southern ocean. This connection has been extensively studied on different time scales, as it plays a crucial role in controlling productivity patterns in the Arabian Sea. Describing the contribution of the ITF to the overall $\delta^{15}\text{N}$ cycle of the Indian Ocean is undoubtedly important, but again, in terms of my understanding, nothing 'new' in the sense of a glaring gap of understanding.

We believe that the referee's points and our arguments are not mutually exclusive. While the Indian Ocean is indeed directly influenced by the Southern Ocean, the ITF also plays an important role in shaping the temperature, salinity, and nutrient content in the more distant regions of the Indian and Atlantic Oceans (e.g. Ummenhofer et al., 2021).

Ummenhofer, C. C., Murty, S. A., Sprintall, J., Lee, T., & Abram, N. J. (2021). Heat and freshwater changes in the Indian Ocean region. *Nature Reviews Earth & Environment*, 2(8), 525-541. doi:10.1038/s43017-021-00192-6

Irrespective of the above statement, their results still constitute the first data set I have seen that convincingly and compellingly shows this connection and its variability over the last 800 kyr. Thus, at the end of my three read-throughs of their manuscript, I am still immensely disappointed by the author's presentation of the significance of this enticing dataset. The lack of willingness to contextualise the complexity within the framework of existing literature is especially baffling, and to any knowledgeable reader, it will be similarly obvious and jarring.

We are glad to hear that the referee considers our dataset to be '*the first dataset I have seen that convincingly and compellingly shows this connection and its variability over the last 800 kyr*'. Regarding the referee's comments on the introduction, please refer to our specific response above. We would also like to emphasize that not every new study, particularly short-format papers, is expected to provide a comprehensive review or discussion of the existing literature. We chose to use only those references required to provide the most salient background rather than an exhaustive review of the topic.

Reviewer #3 (Remarks to the Author):

The study investigates the relative contribution of North and South Pacific in terms of sourcing nitrate to the Banda Sea over the past 800,000 years using nitrate isotopes. Subsequently, these findings are used to infer the fractional contribution of the South Pacific water on the Indonesian Throughflow (ITF) into the Indian Ocean. The study is well-written, and of significant interest to the broader oceanographic community; since it contributes to our understanding of water composition and nutrient transport associated with the Indonesian Throughflow, and the resulting effects on nutrient availability and distribution across the Indian Ocean and on a global scale. The scope of the study and its findings are well-suited for publication in Nature Communications. In my opinion, the methods are well explained, robust and support well the authors' arguments; although I want to highlight that I am not an expert in isotope analysis. However, I do have some minor queries mainly concerning the representativeness of the Banda Sea for the entire ITF. Hence, I recommend some minor revisions as described below prior to publication.

1. Banda Sea and ITF: Throughout the study, the ITF transport and the water passing through the Banda Sea are treated as equivalent. While a significant portion of the ITF outflow through the Ombai and Timor Straits does indeed flow through the Banda Sea, this simplification is not entirely accurate. Observations indicate that approximately 20% of the ITF transits through the shallow Lombok Strait, directly downstream from Makassar and before the flow turns eastward towards the Banda Sea. Therefore, the maximum proportion of the ITF that can be explained by examining the Banda Sea would be 80%. Moreover, recent high-resolution models (Guo et al., 2023) suggest that some water from the Ombai Strait and along the Nusa Tenggara region exits directly into the Indian Ocean before move eastwards towards the Banda Sea, which further reduces this percentage. Additionally, the contribution from the Lifamatola Passage to the Ombai Strait (30% of the total ITF) appears to primarily follow a path along the western boundary of the Banda Sea, rather than traversing its interior. Hence, in my opinion, inferences made from observations within the interior of the Banda Sea are most applicable to the portion of the ITF flowing through the Timor Sea, which accounts for approximately 50% of the total ITF. I suggest that you explicitly highlight and clarify this distinction throughout the text, particularly in the Introduction and conclusions. Specifically, that your discussion regarding the contributions of

South and North Pacific sources to nutrient transport relates to this Timor Sea component of the ITF (or at least a portion of the ITF), rather than the entire ITF.

Guo Y, Li Y, Yang D, Li Y, Wang F and Gao G (2023) Water sources of the Lombok, Ombai and Timor outflows of the Indonesian throughflow. *Front. Mar. Sci.* 10:1326048. doi: 10.3389/fmars.2023.1326048

This is a well-taken point by the referee. The most accurate data measured from different outflow passages of the ITF during the INSTANT campaign (2004-2006) indicate that about 17% of the total volume flows through the Lombok Strait (Sprintall et al., 2009). It follows that 83% of the ITF flows through the Banda Sea, which is the maximum estimate of the ITF volume passing our site. As the referee points out, results from a regional ocean model (no field data) indicate that most of the ITF, when transiting from the Lifamatola Passage to the Ombai Strait, flows along the western Banda Sea (Guo et al., 2023). This translates to maximum 33% of the ITF, leaving 50% of the ITF passing our site (the lowest estimate). However, we note that results from another regional ocean model combined with satellite and in-situ observations indicate a cyclonic circulation of surface waters (upper 500 m) in the Banda Sea (Liang et al., 2019), which means that all the surface waters in the Banda Sea pass through our site. Altogether, the 50% suggested by the referee is the minimum estimate of the ITF component covered by our data, and 83% the maximum. We therefore refer to the ‘majority’ or ‘substantial fraction’ of the ITF in the revised version and added this information to the SI (Text S2).

Liang, L., Xue, H., & Shu, Y. (2019). The Indonesian Throughflow and the Circulation in the Banda Sea: A Modeling Study. *Journal of Geophysical Research: Oceans*, 124(5), 3089-3106. doi:10.1029/2018jc014926

2. Figure 1. To help readers, particularly those unfamiliar with the Indonesian Throughflow, I recommend illustrating the complete ITF pathway in your schematic depiction. Specifically, in Figure 1a, please include the outflows through the Lombok Strait and the Ombai Strait, in addition to the existing outflow through the Timor Sea. This will provide a more comprehensive visual representation.

Done.

3. Lines 102-108, Figures 1b and S1, and use of U1486 to represent the $\delta^{15}\text{N}_{\text{South}}$: U1486 (purple) exhibits significantly different and larger values (lines 96-98) compared to the other 'southern' profiles, falling outside their standard deviation. So I am not sure why you used this record-site to characterise and estimate the fractional contribution of southern vs northern water sources in your equation in line 104 and Figure S4. Maybe I am missing something or I have misunderstood but could you please clarify for this selection. Additionally, could you please provide an alternative estimate of the fractional contribution using a different southern record, or

perhaps the mean of these records. Given the linear combination of $\delta^{15}\text{N}_{\text{North}}$ and $\delta^{15}\text{N}_{\text{South}}$ in the equation in line 104, using the mean and standard deviation of both northern and southern records could provide a mean fractional contribution estimate with a corresponding standard deviation. In my opinion, this approach will strengthen your argument and reduce its dependence on the choice of specific record-sites.

This is another well-taken point by the referee. The $\delta^{15}\text{N}$ record of U1486 is the only Southern Hemisphere-sourced record that extends far enough back in time to serve as a reference for comparison with our Banda Sea record. As mentioned by the referee and outlined in the revised version (lines 107-108 in the clean version of the revised manuscript), the Southern Hemisphere contribution based on this record would be an absolute minimum estimate. By using the 25 kyr stack records to approximate the Northern and Southern Hemisphere source signals, the calculated Southern Hemisphere contribution ranges from 42% to 85% with a mean of 62%, which is consistent with our estimate.

4. Depth of records: This probably stems from my ignorance in terms of using $\delta^{15}\text{N}$, but I was wondering whether the results are sensitive to the water depth of these records. Does the U1486 shallower depth contribute to its observed differences and higher $\delta^{15}\text{N}$ than the rest of the southern records. Similarly, does the considerably deeper depth of MD01-2380 (3232 m) relative to both the northern and southern records affect the overall findings. Given the broader interest in your results, a brief explanation of how water depth might influence $\delta^{15}\text{N}$ analysis and your results (if it does influence them) would be beneficial, particularly for readers less familiar with isotope analysis.

The water depth does not influence the $\delta^{15}\text{N}$ signal per se. Only if water depth reflects the proximity to land, then the $\delta^{15}\text{N}$ might be additionally influenced by varying terrestrial input at different sites. We exclude this potential bias and explain it in the supporting information (Text S1). A potential effect of diagenetic overprint is also discussed and excluded in Text S1.

Reviewer #3 (Remarks to the Author):

As I highlighted in my review of the previous version of the manuscript, the study is of interest to the wider oceanographic community as it contributes to improving our understanding of the Indonesian Throughflow and associated water/nutrient transport into the Indian Ocean. The study's conclusions and authors' arguments are well supported by the analysis and methodology. My previous minor comments have been addressed in the revised version of the manuscript, and I recommend the manuscript be accepted for publication as is.

We again thank the referee for the review and for the positive evaluation of our study.

Reviewer #4 (Remarks to the Author):

Overall remarks

I have been asked to review this manuscript at a time, where a rebuttal already answered to comments from three reviewers. In my review, I will provide my own comments, and include my thoughts on previous comments and their responses in the rebuttal.

Kienast et al. provide new, exciting paleoclimate data which use a unique approach to distinguish northern vs southern water mass sources at their site. I am excited by the nitrogen isotopic approach on multiple cores, and support the publication in Nature Communications, but I have several major points that need clarification before I can support the publication of the manuscript.

We thank the referee for this feedback and for valuing the unique approach of our study. We addressed the referee's comments as outlined below. **Please note that line numbers refer to the clean version of the revised manuscript.**

These include

(a) The current methods section is not well organised and does not include sufficient information on the methodology of the work, especially for a journal like Nature Communications that reaches a wide audience. This makes it hard for any reader that does not have a geochemical background to judge whether the assumptions made in this manuscript are reasonable or not. First, it appears that the authors measured $d_{15}N$ on two cores MD01-2380 and U1486, but the methods only describe steps taken for MD01-2380. Second, the method section "Isotope measurements" does not reference any methodological studies, hence I assume this method is not yet published. In that case, I would expect a more in detail description of how the sedimentary $d_{15}N$ was collected. For example, after homogenisation, how did the samples enter the mass spec? Third, the chapter references a row of cores that also provided samples for bulk $d_{15}N$ analysis, but these cores are not really referred to in the main manuscript. More information helping the reader to piece together the materials & methods would be very helpful. Fourth, both chapters "Biogenic opal" and "Alkenone analysis" do not clarify what core material was used. It is unclear what core was used for what

analysis. Similarly, it is unclear what cores were used for the spectral analysis. U1486 is referenced in the text, but the spectral analyses do not appear in Figure 3. Fifth, it appears the authors stacked a suite of records in Fig. 1b. However, the description of this stacking process is very limited. Questions remain regarding why was 500 years chosen as the interpolation time step? What exactly does “time dependent mean” refer to? I suggest the authors include a mathematical formula in the supplement. How exactly was the error envelope plotted in Fig 1b calculated? What errors are included in the error margins? Does it include age model errors? Several stacking approaches are published, did the authors choose their own approach or used an already published approach?

The “Methods” section has been revised to improve its clarity.

First: As outlined in the main text of the manuscript, our study presents a newly generated 800 kyr $\delta^{15}\text{N}$ record of MD01-2380 (lines 60-61). To address the origin of the isotopically enriched nitrate in the Banda Sea, we compare this record to previously published and new $\delta^{15}\text{N}$ records from the WEP for the last 25 kyrs (lines 77-79, Fig. 1b; Fig. S1), from the northern WEP offshore Mindanao for the last 160 kyr (site MD06-3067), and from the southern WEP offshore Papua New Guinea for the last 800 kyrs (site U1486) (lines 79-81). The $\delta^{15}\text{N}$ record of U1486 was originally published by Lambert et al. (2022). This is referenced in the main text (line 81), as well as in the captions of Fig. 2, and Table S1. We have additionally added the missing information to the Methods section (lines 234-245) to clarify the origin of all the available individual $\delta^{15}\text{N}$ records.

Second: The methods for measuring the $\delta^{15}\text{N}$ of organic matter are well-established, but we have added the following text and a reference to the method to comply with the referee’s request:

Sedimentary $\delta^{15}\text{N}$ ($\delta^{15}\text{N}_{\text{sample}} = [(\text{N}^{15}/\text{N}^{14})_{\text{sample}} / (\text{N}^{15}/\text{N}^{14})_{\text{reference}} - 1] * 1,000$) was analyzed on ca. 60 mg of freeze-dried sediment, homogenized in an agate mortar, packed and enclosed within a tin capsule, placed in a carousel, and combusted to N_2 for N isotopic analysis (Verardo et al. 1990) (lines 254-255).

Third and fourth: We have added a “Material” paragraph to clarify that samples of MD01-2380 were used for isotope analysis and for biogenic opal and alkenone measurements (lines 234ff). We have also added missing information about the core material to the “Biogenic opal” and “Alkenone analysis” sections (lines 264 and 273, respectively). In the main text, we outline that “we compare $\delta^{15}\text{N}$ of site MD01-2380 in the Banda Sea to previously published and new $\delta^{15}\text{N}$ records from the WEP for the last 25 kyrs (stack records) (lines 77-79). For more details, we refer the reader to the methods section. Here, we now list the WEP cores that we used to generate new WEP $\delta^{15}\text{N}$ records for comparison with MD01-2380. An overview about the stack records is additionally provided in Table S1.

We calculated power spectra of the $\delta^{15}\text{N}$ record of MD01-2380 (line 129), of the U1486 - MD01-2380 $\Delta\delta^{15}\text{N}$ (lines 170-171) and of the opal content and alkenone concentration of MD01-2380. We added this information in a summarized form to the Methods (lines 298-299). In reply to the other comments of the referee, the revised manuscript also includes the spectra of the $\delta^{15}\text{N}$ records of U1486 and MD06-3067 (Fig. S6). Previously missing

information, which core has been used for which analysis has been added (e.g., lines 234ff, caption of Fig. 3).

Fifth: We used a commonly applied approach to generate the Northern and Southern Hemisphere stack records: Prior to the calculation of the stack records, all time series were linearly interpolated to even time steps of 500 yrs. The selected time step of 500 yrs matches the average resolution of the stack records. For some cores, this results in a slightly higher resolution than the original records (or simply “slight oversampling”). However, the resampling does not affect our results and their implications. A comparison of the original records and the interpolated records is shown in Fig. S1. We then calculated the time-dependent mean and the corresponding standard deviation. The time-dependent mean is the mean $\delta^{15}\text{N}$ of the individual records included in the stacks at each time step. This information is now added in the method section (lines 320-321). We did not include age model uncertainties in the error envelope, as age model uncertainties do not affect the main result, which is a substantially different isotopic composition of NH and SH sourced waters during the past 25 kyrs. Please note that in this study, we refrain from discussing timescales that might be affected by age model uncertainties.

(b) I would like to see a more careful approach regarding age models. First, the “age model” section in the methods clarifies that core MD01-2380 was tuned to U1486 using planktic foraminifera. This is a somewhat unusual method, as usually methods call for benthic foraminifera to be tuned to each other and ultimately to the orbital parameters. Maybe the authors can include a sentence or two on what age model errors are expected using this approach. Second, it is currently unclear whether the foraminifera $\delta^{18}\text{O}$ record of U1486 is planktic or benthic. I assume planktic? It is important to know whether the age model compares “apples with apples”. Third, there is no description of how the age model for U1486 is constructed, or any of the other cores. What about the age models for the other cores listed above in chapter “Isotope measurements”? Fourth, how did the authors deal with age model differences when stacking the records in Fig 1b? How did they incorporate age model errors in the error envelope?

First and second: Our age model is based on radiocarbon dates and on the alignment of the planktic $\delta^{18}\text{O}$ record of MD01-2380 to the *G. ruber* $\delta^{18}\text{O}$ record of U1486. This is now clarified in lines 285-287. The $\delta^{18}\text{O}$ records are also presented in Fig. S2. Please note that the age model for Site U1486 was constructed using a ^{14}C date, x-ray fluorescence $\ln(\text{Ti}/\text{Ca})$ correlated to nearby Core MD05-2920, and by aligning benthic $\delta^{18}\text{O}$ to the LR04 benthic stack (Lambert et al., 2022).

Third: The age model of U1486 was adopted from Lambert et al. (2022). We added this information in lines 287-288. Hence, for this age model, as well as for the age models of MD06-3067 and the records that are included in the NH and SH stacks, we refer the reader to the original publications. We added this information in lines 294-295. As outlined above, age model uncertainties of the records that are included in the NH and SH stacks do not affect the main result, which is a substantially different isotopic composition of NH and SH sourced

waters. As mentioned above, we do not discuss timescales that might be affected by age model uncertainties.

(c) The discussion on ITF mixing ratios, as well as the use of the term “ITF”, vs. “Banda Sea” is confusing. I would like clarity for the reader in terms of what the authors mean by “ITF”. I suggest the authors move away from writing about the “ITF” and focus on their study site which is in the Banda Sea. The mixing of water masses is currently discussed in the introduction as well as the methodology section as “Text S1, S2” etc. I find this structure very unintuitive, and not supportive for the manuscript. I suggest that the authors take the information out of the “Texts” and include them in the manuscript. Kindly describe the water masses and their percentages of total water as they appear in the Banda Sea, and then in the same fashion as they appear in the east Indian Ocean, where they can be called the “final” ITF. This should include sources that may not channel via the Banda Sea. I suggest that the authors then do a back-of-the-envelope quantification of the fraction of southern-sourced waters in the Banda Sea and the “final” ITF in terms of % of modern volume flow. Then this quantification can be taken back in time. Like this, the reader will be able to follow how much the contribution impacts the records at the Banda Sea core site, and subsequently the “final” ITF.

We thank the reviewer for raising this point. We agree that clarity is important, but we believe that retaining the term “ITF” is essential because the Banda Sea is not an isolated basin. It is located in the heart of the ITF and, as outlined in the manuscript, the majority (up to 83%) of the ITF circulates through the Banda Sea, making it a key reservoir and mixing zone for Pacific waters. Our study uses the Banda Sea record to infer changes in the ITF nutrient composition and hemispheric sources through time, with implications for regional nutrient transport and global circulation. Removing the term “ITF” would disconnect our findings from this broader context.

The suggestion to move Texts S1 and S2 to the main manuscript conflicts with the feedback of referee 2, who suggested that the information currently provided in Text S1 and initially included in the main text (discussion) would be better suited in the supplementary part of the manuscript. To balance these perspectives, we added a few key details about the main water masses feeding into the ITF to the introduction (lines 23-27), while keeping a more comprehensive description of modern oceanography in Text S2 to avoid distraction from the manuscript’s focus. In the main text, we only discuss water masses and circulation features that directly influence thermocline nutrient dynamics and export production, which is the key focus for the paleoceanographic interpretation.

We appreciate the reviewer’s suggestion to provide more detail on water masses and their relative contributions.

In response to modern and past volume transport, we would like to emphasize that we are assessing the relative contribution of Northern vs. Southern sourced nitrate (via $\delta^{15}\text{N}$) to document the admixture of thermocline waters in the Banda Sea, i.e. the layer that provides the nutrients to fuel export production rather than deeper water masses passing through. We did a back-of-the-envelope quantification of the fraction of the relevant southern-sourced

waters in the Banda Sea and the final ITF, as suggested, and included this in the main text (lines 108-128).

(d) After having noted my own comments, I read the rebuttal and noticed that many of my comments already appeared in the previous round of reviews. For example, previous reviewers already asked for a more detailed method section/supplement, and a more nuanced discussion on the ITF vs the Banda Sea. I would like to encourage the authors to respond to reviewer comments with more enthusiasm and improve their manuscript. I also believe some comments have not been addressed. First, the request for a more in-depth discussion of the spectral analysis results. I encourage the authors to satisfactorily discuss what implications their results have on larger scale changes, such as sea level variability. The authors state that any interpretation would be “purely speculative”, but I respectfully disagree. In my view it is quite an interesting finding that the authors don’t see a strong 100kyr cycle given the sea level dynamics in the region over the Pleistocene (e.g. Nuber et al., 2023). What hinders this signal to appear in the Banda Sea? Second, I do not follow the authors’ response on the C/N ratio. What does the attached plot show? What samples are plotted here? How exactly are the authors justifying their method? I suggest including this plot and this method validation into the supplement and providing much more detail. Third, the authors claim in the rebuttal that the $\delta^{15}\text{N}$ in the Banda Sea is not depth dependent, only as distance to land. Surely degree of stratification would influence the $\delta^{15}\text{N}$ as different water masses enter the Banda Sea? Is that not one of the concepts that the authors use to calculate the amount of southern-sourced water?

First: A number of studies from the Makassar Strait and the main outflow passage of the ITF (Timor Sea) indicate a shift from a more surface flow during glacials to a thermocline-dominated flow during interglacials, as a result of rising sea-level and inundation of the Java Sea (e.g. Ding et al., 2023; J. Xu et al., 2006). According to these studies, the relatively fresher waters of the Java Sea induce a “freshwater plug” south of the Makassar Strait during interglacials and inhibits a southward flow of surface waters there, and facilitate the thermocline flow of the ITF, akin to its modern seasonal variation (Gordon et al., 2003; Sprintall et al., 2009). However, a change from surface to thermocline flow of the ITF would not affect our $\delta^{15}\text{N}$ record from the Banda Sea, as this is derived from the thermocline (surface waters are nutrient-depleted) and changes only if the relative contribution of the Northern and Southern Hemisphere source waters would change. Moreover, the net contribution of Java Sea/Karimata Strait to the ITF is only about 0.8 Sv of surface flow (Sprintall et al., 2019; T. F. Xu et al., 2021). The Karimata and Lombok Straits are the only shallow passages of the ITF that are affected by changing sea-level on glacial-interglacial scale. They hardly affect the thermocline conditions, and are unrelated to the $\delta^{15}\text{N}$ composition, in the Banda Sea. In summary, although sea-level changes might be capable of changing the uppermost vertical profile of the ITF on glacial-interglacial timescales, they did not affect the relative contribution of the Northern and Southern Hemisphere source waters to the ITF. The few available studies that suggest a change in the relative contribution of the Northern and Southern Hemisphere to the ITF predate our records (e.g. Auer et al., 2019; Petrick et al., 2019). Notably, major sea-level driven changes in the relative contribution of

the Northern and Southern Hemisphere sources to the ITF relate to tectonic reorganizations (e.g. Auer et al., 2019). We added a paragraph on this topic (lines 201-217).

Second: C/N ratios and $\delta^{13}\text{C}_{\text{org}}$ are commonly used to verify the marine origin of bulk organic matter, because C/N ratios and values of $\delta^{13}\text{C}_{\text{org}}$ from marine and terrestrial sources are remarkably different. While marine sources typically have C/N ratios between 4 and 10, terrestrial plants have C/N ratios above 20. Typical $\delta^{13}\text{C}$ values of marine organic matter are between -22 and -20 ‰ (e.g., Meyers, 1994). The average C/N ratio of 8.6 and average $\delta^{13}\text{C}_{\text{org}}$ of -20.9 ‰, measured on multiple MD01-2380 samples, thus clearly indicate a marine origin of the bulk organic matter. We added this explanation to the method validation that is already included in text S1 of the manuscript to improve the comprehensibility of the manuscript for a broader audience.

Third: In the rebuttal we refer to the water depth of the core sites and argue that it does not influence the $\delta^{15}\text{N}$ signal per se. Only if water depth reflects the proximity to land (meaning when the depth of the core sites correlates with their $\delta^{15}\text{N}$ signal), then the $\delta^{15}\text{N}$ might be additionally influenced by varying terrestrial input at different sites. This does not refer to water mass stratification, which is discussed in the main text as follows: “Assuming that there is a depth stratification to the Northern and Southern Hemisphere source of the ITF waters as inferred from modern T/S observations in the Banda Sea², deeper mixing would be tapping into more Southern-Hemisphere sourced waters, resulting in enriched $\delta^{15}\text{N}_{(\text{nitrate})}$. Such a deep mixing is rather likely during austral winter and spring, i.e., during periods of low precession³⁷” (lines 134-135).

Line by line comments

Main text:

Line 83: Kindly include a more in-depth discussion on the endmembers. What are the endmember $\delta^{15}\text{N}$ values for these different water masses? Are they stable through time? Are they distinct water masses in the Banda Sea? Or vertically mixed already? A more nuanced discussion of the ITF is needed.

Modern day endmember values (published in Lehmann et al., 2018) are discussed in lines 46-53, describing the offset between northern and southern thermocline values and the processes involved leading to these differences in isotopic signature. In the revised manuscript, we added endmember values to lines 49 and 51-52 for clarification.

Line 107: the sentence states “Offset between southern hemisphere record averages 2.3permil... Figure 2”. I am slightly confused, as the previous sentence in line 105 already quotes 2.3permil for the difference in the northern WEP stack. In Fig 1, the southern hemisphere stack is clearly less offset than the NH stack. In Fig 2, why do the authors calculate a $\Delta\delta^{15}\text{N}$? Presumably the authors want to reference Fig 2c, rather than the entire Fig 2? Are the authors referring to the average value of 2c? If so, it would make sense to briefly explain why this matters, what it expresses, and include a line to show where the average is.

The offset between our Banda Sea $\delta^{15}\text{N}$ record and the northern WEP stack averages about 2.7 ‰. We corrected this typo and additionally provide the information that the offset between the Banda Sea record and the SH stack averages about 1.6 ‰ during the last 25 kyrs. In the manuscript, we also provide the offset between the SH record at site U1486 and our Banda Sea record, which is about 2.3 ‰ during the past 800 kyrs. The $\Delta\delta^{15}\text{N}$ record (U1486 – MD01-2380) is shown in Fig. 2c and now correctly referenced (lines 92-93). As outlined in detail in the manuscript (particularly lines 100-128), it shows the permanent and significant admixture of Southern Hemisphere nitrate to the ITF flowing through the Banda Sea during the past 800 kyrs.

Line 113: Can the authors reference this sentence. What modern observations are we talking about? Are they plotted anywhere?

As referenced in line 99, we are referring to modern data presented in Lehmann et al. (2018).

Line 127: Not sure what the authors mean by “absolute minimum estimate”

We assess that the “Southern Hemisphere contribution ranged from less than 10% to more than 90%, with an average of slightly above 50 % for the past 160 kyrs (Fig. S4)” (lines 105-106). This is a minimum estimate, because the $\delta^{15}\text{N}$ signal at site U1486 is at the high end of the alleged Southern Hemisphere $\delta^{15}\text{N}$ source (lines 114-115), above the Southern Hemisphere stack values.

Line 138: These claims are hard to see in the provided plots. Maybe include shaded bands to highlight precession or obliquity pacing for the eye. Or include cross-plots of orbital parameters and $\delta^{15}\text{N}$ to show these correlations visually.

We did as suggested and included shaded bands to highlight precession and obliquity pacing in Fig. S5.

Line 146: how do Figures 3, S3 support this statement? Fig. 3 only shows the spectral analysis of core MD01-2380, and Figure S3 does not show any spectral analysis. Is there another figure that might be leveraged to show this more clearly?

We added an additional Figure that shows the power spectra of the $\delta^{15}\text{N}$ records of MD06-3067 (Northern Hemisphere) and U1486 (Southern Hemisphere) (Fig. S6).

Line 159: It appears that there is a water mass here which provides a substantial amount of water for the ITF, but might be left out in the authors’ proxy, due to the lack of nutrients. Have the authors considered this when calculating their southern-sourced water contribution? As stated above, I recommend the authors do a calculation where they clearly state what the

southern-sourced water contribution is in Sv, and where they account for all other sources in Sv too.

Our study documents the fractional contribution of subsurface / thermocline nitrate in Northern and Southern Hemisphere source waters and thus, the admixture of these waters in the ITF through time. In this comment, the referee is referring to eastern margin surface waters that may constitute a component of ITF surface waters. In fact, the modern ITF is thought to transport mainly thermocline waters (e.g. Sprintall et al., 2014). See also below our back-of-the-envelope calculation in response to referee's last comment.

Line 186: I am unsure how Figure 3 supports this statement. Figure 3 does not show spectral analyses from U1486.

Here, we reference the original publication of the U1486 $\delta^{15}\text{N}$ record (Lambert et al., 2022), which discusses that the obliquity signal in this record is not dominant during the past 800 kyr (after the Mid Pleistocene Transition). In the revised version of the manuscript, we additionally include the power spectrum of the past 800 kyr of the record in Fig. S6. It confirms that the influence of obliquity on the $\delta^{15}\text{N}$ at site U1486 is not dominant.

Line 194: I do not see that the two records are particularly similar across the entire record. I am surprised by their differences. That said, these are all "subjective by eye" statements. I suggest the authors use some statistics to explore how similar/different these records are. For example, the authors can use a sliding window and compute a correlation coefficient. Or calculate differences to highlight regions of similarity vs difference. I am encouraging the authors to explore their own methods depending on what it is they want to show.

We specified our statement to clarify that we refer to the precession variability of the records (line 187-188). Specifically, both records indicate a higher production during periods of high precession. In Fig. 3, we show power spectra of both records. The records of both alkenone concentrations and percent concentrations of biogenic opal peak at frequencies that correspond to the precession cyclicity.

Line 198: I have difficulty seeing the correlation between orbital parameters and the records. I suggest re-plotting in a different way (ie using cross-plots or other methods as suggested above).

This has been addressed in response to the referee's comment on line 138.

Line 200: reference a plot, where do we see that peak primary production occurs during austral winter?

This statement refers to Beaufort et al. (2010). This study is also referenced in line 192.

Line 205: reference a figure, Where do we see that obliquity is only evident in biogenic opal not alkenones? What core are we talking about?

We added a reference to Fig. 3b, which shows the power spectra of the opal and alkenone records. We now explicitly mention that we refer to the biogenic opal and alkenone records of MD01-2380 (line 199). In our study, we only present biogenic opal and alkenone records of MD01-2380. We slightly revised the method sections (“Material”, “Biogenic opal” and “Alkenone analysis”) to clarify this.

Line 210: define “substantial”. I am looking forward to reading a number in Sv or as % of total Sv here.

To explore the potential magnitude of hemispheric volume transport, we did a simple back of the envelope calculation (see Methods for details). Considering only the shallow throughflow (0-300 m), our approximation suggests that Southern Hemisphere sources contribute up to 6.8 Sv, with an average of ~2.0 Sv to the ITF outflow through Timor Passage and Ombai Strait (corrected total of ~8.0 Sv; Methods) over the past 160 kyrs (Fig. S4).

Reviewer #5 (Remarks to the Author):

Review of “Significant Southern Hemisphere contribution to the Indonesian Throughflow over the last 800,000 years” by Kienast et al.

A few pages into this paper I realized that I should perhaps not have agreed to review it. Reviewers 1-3 clearly have more expertise in tracers and in Pacific Ocean water masses than I do. But this makes me more of a general reader, and since articles from this journal are intended for a broad audience, I might be able to contribute something.

I confess to being initially surprised that the conclusion that a SH source for the Throughflow should be novel: as Reviewer 1 points out, Godfrey’s island rule predicts a net northward flow east of Australia of about 15 Sv.. This transport is the net Sverdrup transport minus the transport of the western boundary current along Australia’s east coast. So really ALL the ITF Throughflow has a SH origin, and the debate seems to be the partitioning between waters that take a more or less direct route into the Indonesian Archipelago and those that have been seen substantially modified as a result of residence in the NH. The authors attempt to quantify SH influence over the past 800,000 years by comparing the history of $\delta^{15}\text{N}$ from a sediment core taken in the central Indonesian Archipelago to cores taken from locations of established NH or NW origin. Reviewers 1 and 3 are favorable, and all three reviewers acknowledge the importance of the data. Review 2 is unhappy with the original submission, feeling that the novelty of the findings were overstated, that insufficient coverage was given to recent literature, and that elements of presentation was poorly framed. My reading of the revised

version suggests that sufficient caution is exercised in stating conclusions, and that recent literature is at least widely cited. I do think that despite the improvements the paper is still going to be challenging for an outsider like myself, and I also have a few questions about the analysis, but my impression is that the work is nearly publishable.

We thank the referee for the effort to review our manuscript and for the positive evaluation of our manuscript and revisions. We address the specific comments as outlined below. **Please note that line numbers refer to the clean version of the revised manuscript.**

1) It is well known that the Lombok and other passages that drain the Indonesian Archipelago undergo frequent flow reversals as a result of Kelvin waves of Indian Ocean origin. Murray and Arief (Nature, 1988, Fig. 3) show examples. Whether the resulting northward inflows of water of Indian Ocean origin make their way into the Banda Sea, I do not know. Is this a consideration?

This is a well-taken point by the referee. Measurements, for instance from the INSTANT campaign, showed that the passage of Kelvin waves through the Ombai Strait permits Indian Ocean water to penetrate into the Banda Sea (Sprintall et al., 2009). However, it is unknown how that affects the interior of the Banda Sea. Any discussion would thus be purely speculative here.

2) Is anything known about changes in Pacific surface wind field over the past 800,000 years? This could potentially effect Godfrey's estimate of the net northward transport.

With this comment, the referee addresses a previous comment of J. R. Toggweiler after the original submission of the manuscript. The short answers are no and yes, respectively. The overall structure and direction of the surface wind field are controlled by the Coriolis force and land-ocean configuration, which would not change on timescales relevant for our study. Their strength varies though, for instance in response to orbital forcing. However, reconstructing wind fields in the past has always been difficult due to the lack of an undisputed proxy, and so far, only qualitative. Means even if we knew the direction of past changes, we would not know by how much, making their translation to Sv transport untrustworthy.

3) I don't agree with your response to Reviewer 2's questions about G-IG sea level variation. It's a very natural question the average reader will have since sea level changes could have interesting effects on the way that the ITF is routed. And, if the ITF is really "choked", this may have upstream ramifications. I don't think it would "confuse the reader" to mention this, perhaps as just a caveat. And perhaps significant to point out that you do not see evidence in the core samples of such variability (if I understand that correctly).

This was also a comment by Referee #4. We included a short discussion on this topic in lines 201-217.

Other points:

Line 58. Surface temperature is certainly not conservative. Neither are nutrients.

We agree. However, in contrast to sea surface temperature, $\delta^{15}\text{N}$ is barely modified by contact with the atmosphere.

Line 78 refers to site MD01-2380 as labeled by a star in Fig. 1a, but the figure has the star labeled as MD80.

As the referee is correctly noting, we use abbreviations of the core names within the figure to keep it clear. This is now clarified in the figure caption.

Line 109. If you are going to refer to the U1486 curve in Fig. 1b. it would be helpful to label it.

The records, including U1486, are color-coded and labelled in Fig. 1a, as well as in Fig. S1.

Line 119. You use only U1486 to represent $\delta^{15}\text{N}_{\text{South}}$, apparently because you feel it is best correlated with MD01-2380 in Fig. 1b. MD20 is geographically quite close to U1486 but is not used, which raises some questions in my mind. This also relates to what is written later on lines 125-127. I suppose that, just for comparison, you could have done a second calculation using the average of the SH sources to represent $\delta^{15}\text{N}_{\text{South}}$.

We used U1486, as it is the only Southern Hemisphere-sourced record that extends far enough back in time to serve as a reference for comparison with our Banda Sea record. Core MD05-2920 (MD20) covers the past 400 kyr, however the $\delta^{15}\text{N}$ record of this core only covers the past ~60 kyr. This record is considered in the SH stack. As demonstrated in Fig. 1b, the offset between U1486 and MD01-2380 is larger than the offset between MD05-2920 and MD01-2380. In the revised version, we also provide an assessment of the SH contribution to the ITF that is based on the comparison of the SH stack record and MD01-2380 (lines 90-91).

References

- Auer, G., De Vleeschouwer, D., Smith, R. A., Bogus, K., Groeneveld, J., Grunert, P., . . . Henderiks, J. (2019). Timing and Pacing of Indonesian Throughflow Restriction and Its Connection to Late Pliocene Climate Shifts. *Paleoceanography and Paleoclimatology*, 34(4), 635–657. 10.1029/2018pa003512
- Beaufort, L., van der Kaars, S., Bassinot, F. C., & Moron, V. (2010). Past dynamics of the Australian monsoon: precession, phase and links to the global monsoon concept. *Climate of the Past*, 6(5), 695–706. 10.5194/cp-6-695-2010
- Ding, X., Bassinot, F., Pang, X., Kou, Y., & Zhou, L. (2023). Heat transport processes of the Indonesian Throughflow along the outflow pathway in the eastern Indian Ocean

- during the last 160 Kyr. *Paleoceanography and Paleoclimatology*, 38(e2023PA004620). 10.1029/2023PA004620
- Gordon, A. L., Susanto, R. D., & Vranes, K. (2003). Cool Indonesian throughflow as a consequence of restricted surface layer flow. *Nature*, 425(6960), 824–828. 10.1038/nature02038
- Lambert, J. E., Gibson, K. A., Linsley, B. K., Bova, S. C., Rosenthal, Y., & Surprenant, M. (2022). Equatorial Pacific bulk sediment $\delta^{15}\text{N}$ supports a secular increase in Southern Ocean nitrate utilization after the mid-Pleistocene Transition. *Quaternary Science Reviews*, 278. 10.1016/j.quascirev.2021.107348
- Lehmann, N., Granger, J., Kienast, M., Brown, K. S., Rafter, P. A., Martínez-Méndez, G., & Mohtadi, M. (2018). Isotopic Evidence for the Evolution of Subsurface Nitrate in the Western Equatorial Pacific. *Journal of Geophysical Research: Oceans*. 10.1002/2017jc013527
- Meyers, P. A. (1994). Preservation of elemental and isotopic source identification of sedimentary organic matter. *Chemical Geology*, 114(3), 289–302. [https://doi.org/10.1016/0009-2541\(94\)90059-0](https://doi.org/10.1016/0009-2541(94)90059-0)
- Petrick, B., Martínez-García, A., Auer, G., Reuning, L., Auderset, A., Deik, H., . . . Haug, G. H. (2019). Glacial Indonesian Throughflow weakening across the Mid-Pleistocene Climatic Transition. *Sci Rep*, 9(1). 10.1038/s41598-019-53382-0
- Sprintall, J., Gordon, A. L., Wijffels, S. E., Feng, M., Hu, S., Koch-Larrouy, A., . . . Setiawan, A. (2019). Detecting Change in the Indonesian Seas. *Frontiers in Marine Science*, 6. 10.3389/fmars.2019.00257
- Sprintall, J., Wijffels, S. E., Molcard, R., & Jaya, I. (2009). Direct estimates of the Indonesian Throughflow entering the Indian Ocean: 2004–2006. *Journal of Geophysical Research: Oceans*, 114(C7). 10.1029/2008jc005257
- Xu, J., Kuhnt, W., Holbourn, A., Andersen, N., & Bartoli, G. (2006). Changes in the vertical profile of the Indonesian Throughflow during Termination II: Evidence from the Timor Sea. *Paleoceanography*, 21(4). 10.1029/2006pa001278
- Xu, T. F., Wei, Z. X., Susanto, R. D., Li, S. J., Wang, Y. G., Wang, Y., . . . Fang, G. H. (2021). Observed Water Exchange Between the South China Sea and Java Sea Through Karimata Strait. *Journal of Geophysical Research: Oceans*, 126(2), e2020JC016608. <https://doi.org/10.1029/2020JC016608>

Reviewer #4 (Remarks to the Author):

Overall remarks

As previously stated, Kienast et al. provide an exciting study that now clearly disentangles the southern from northern nutrient sources to the ITF using nitrogen isotope tracers. I congratulate the authors on their new manuscript that now answers all my comments and, in my opinion, reads logically to a wide audience. The results and hypotheses are clearly stated, and the data is logically presented. I am pleased to see the fleshed-out version of the methods, which is now also much easier accessible. Some very minor comments below, but ultimately, I am happy to see this manuscript published in its current form.

Warm regards,

Sophie Nuber

We thank Sophie Nuber for her review and for the positive evaluation of our study. We addressed the minor comments as outlined below. Line numbers refer to the clean version of the manuscript.

Minor comments (lines refer to the clean version):

Line 71: The authors reference Fig. 2a, I believe they want to reference the MD01-2380 record which would be Fig 2b?

Yes. We correct the reference.

Line 164: This hypothesis is very compelling, however it left me thinking: how can the authors disentangle north from south, if both are ultimately driven by the same southern signal? The authors describe well that the northern Hemisphere WEP “reflects” (line 162 – 164). However, it may be beneficial to have one sentence here clearly stating what the southern signal reflects. I presume it reflects only the nutrient-rich southern signal, without the overprints experienced by the north?

Yes. We explain the southern signal in lines 158-167.

Line 287: At risk of being pedantic, it would be useful for an interested reader (like myself) to have a brief statement that the age model of U1486 is based on alignment of benthic $\delta^{18}\text{O}$ to LR04.

We added a statement that the age model of U1486 is It is based on a ^{14}C date at the core top, on the alignment of x-ray fluorescence data to a nearby core, and on the alignment of benthic $\delta^{18}\text{O}$ to the LR04 benthic stack (lines 296-299).

Line 324: It would be beneficial to see the modern values included in Figure 1 as symbols for both stacks.

We included modern $\delta^{15}\text{N}$ values in Figure 1.

Figure S5: The figure attached separately to the upload system seems to have cut-off the precession axis. From Figure 2 I assume that precession is plotted with an inverted y-axis? I suggest making a short note of the inverted axis in the figure caption.

We fixed the axis in Fig. S5 and added a short note of the inverted axis to the figure caption.

Line 363: I noticed the additional line in the Acknowledgements. I send the authors my condolences and look forward to seeing this manuscript published.

Thank you.

Reviewer #5 (Remarks to the Author):

The authors have addressed my comments to my satisfaction. I am ok with publication of the article. Larry Pratt

We again thank Larry Pratt for his review and for the positive evaluation of our study.